# AgentSelect: Benchmark for Narrative Query-to-Agent Recommendation

Yunxiao Shi [1]   Wujiang Xu [2]   Tingwei Chen [1]   Haoning Shang [3]   Ling Yang [4]   Yunfeng Wan [5]   Zhuo Cao [6]
Xing Zi [1]   Dimitris N. Metaxas [2]   Min Xu [1]

## Abstract

LLM agents are rapidly becoming a practical interface for task automation, yet selecting suitable deployable configurations remains underexplored. Existing LLM leaderboards and tool/agent benchmarks evaluate components in isolation and are fragmented across tasks, metrics, and candidate pools, leaving a critical research gap: there is little query-conditioned supervision for learning to recommend end-to-end compositional agent configurations. We introduce AGENTSELECT, a benchmark that reframes agent selection as narrative query-to-agent recommendation over capability profiles. AGENTSELECT systematically converts heterogeneous evaluation artifacts into unified positive-only interaction data, comprising 111,179 queries, 107,721 deployable agents, and 251,103 interaction records from 40+ sources across LLM-only, toolkit-only, and compositional agents. Our analyses reveal a shift from dense head reuse to long-tail, near one-off supervision, where ID-based interaction methods become fragile and reliable recommendation increasingly requires content-aware intent-to-capability matching. We validate the synthesized supervision through counterfactual, ablation, and show practical transferability on the MuleRun agent marketplace and a small-scale end to end Agno deployment study. Overall, AGENTSELECT provides the first unified data and evaluation infrastructure for agent recommendation, which establishes a reproducible foundation to study and accelerate the emerging agent ecosystem. The resources are available at [1].

[1]University of Technology Sydney [2]Rutgers University [3]https://github.com/shanghaoning2019 [4]The University of Sydney [5]The University of New South Wales [6]The University of Queensland. Correspondence to: Yunxiao Shi <Yunxiao.Shi@student.uts.edu.au>, Min Xu <Min.Xu@uts.edu.au>.

*Proceedings of the 43rd International Conference on Machine Learning*, Seoul, South Korea. PMLR 306, 2026. Copyright 2026 by the author(s).

[1]https://github.com/Ancientshi/AgentSelect

## 1. Introduction

Modern systems increasingly pair large language models (LLMs) with external tools and execution logic to form AI agents that autonomously carry out complex, multi-step tasks. Over the past two years, most real-world impact has come from carefully engineered, product-grade agents built for specific workflows, such as NotebookLM (Google) and V0 (Palmer). Yet a natural next step for broad adoption is zero-code, on-demand agent creation: enabling non-experts to instantiate lightweight, customized agents (Jiabin Tang, 2025) that solve the narrative queries they pose at the moment of need. The recent success of Ant Group's LingGuang (Ant Group, 2025) hints at this future: it can generate an interactive mini-app from a single conversational instruction, suggesting a path toward massively democratized task automation.

However, realizing this on-demand vision requires more than making agents easier to build; it also requires making them easier to select, configure, and deploy for each task. This challenge appears in two emerging application scenarios. First, in end-user agent assistants built on lightweight agent frameworks and harnesses such as Agno, LangGraph, and OpenClaw, the system should move beyond exposing manual LLM/tool toggles, and instead compose or retrieve an effective agent configuration on demand for users' narrative request. Second, in agent marketplace platforms such as MuleRun (MuleRun), users face a catalog of hundreds of specialized agents and must identify which one is most suitable for their task. Across both scenarios, this abundance of compositional choices creates a practical dilemma: users and practitioners now face a rapidly expanding configuration space, yet there is little principled guidance for deciding which agent or configuration should be selected for a given narrative task.

This motivates a concrete research question: *given a natural-language query and a large catalog of candidate agents, how can we rank agents by expected utility?* Existing leaderboards and benchmarks provide valuable evidence about individual LLM capabilities (Beeching et al., 2023; Fourrier et al., 2024; Suzgun et al., 2023; Hendrycks et al., 2021a; Wang et al., 2024b; Wu et al., 2025; Ma et al., 2025; Mialon et al., 2024) and tool-use performance (Ye et al., 2025;

Huang et al., 2024; Guo et al., 2024; Qin et al., 2024a; Li et al., 2023; Hao et al., 2025; Luo et al., 2025; Fei et al., 2025), but they operate primarily at the component level. Such evidence can help narrow down candidate models or tools, but it does not yield *query-conditioned preferences over complete configurations*—the supervision signal needed to learn a recommender. Moreover, these evaluation artifacts are inherently heterogeneous, varying in tasks, metrics, protocols, and candidate pools, and are therefore often consumed only for benchmark-specific diagnosis rather than being unified and reused as standardized supervision for configuring agents. We bridge this gap by proposing a *query-to-agent recommendation* task and introducing AGENTSELECT, a benchmark and dataset that *standardizes heterogeneous evaluation artifacts* into *query-conditioned supervision* for learning to rank agent configurations.

AGENTSELECT operationalizes this task by ❶ treating each candidate agent as a deployable capability profile $(M, T)$ with an executable YAML specification, and by ❷ unifying and synthesizing supervision signals across model-only, tool-only, and compositional settings into a positive-only query–agent interaction benchmark. This design yields a consistent training and evaluation interface for learning rankers that map narrative intent to configuration-level capability at large scale. ❸ We provide extensive baselines and analyses that reveal a regime shift from dense query–agent reuse to long-tail, near one-off supervision. Under this shift, ID-based interaction modeling works well in dense regimes with repeated query–agent interactions, but becomes fragile in long-tailed settings, where positives are sparse and validation increasingly involves cold-start queries or agents. Meanwhile, neural content-aware rankers, especially TwoTower with explicit TF-IDF features, remain highly competitive in agent remmendation. We also show that stronger and in-domain-tuned embedding retrievers improve long-tail capability alignment but limited in the ID-driven reuse regime. ❹ We further validate the synthesized pseudo-positives via learnability tests and counterfactual capability edits, and show that models trained on AGENTSELECT transfer to real-world agent marketplaces and align with end-to-end execution performance of deployed agents—supporting the external validity and practical significance of the benchmark. The overall framework—covering benchmark construction, recommender training/validation, and practical deployment—is summarized in Figure 1.

## 2. Related Work

### 2.1. LLM Evaluation

*LLM QA leaderboards* (e.g., the Open LLM Leaderboard (Beeching et al., 2023; Fourrier et al., 2024), built on suites such as MMLU/BBH/MATH/MUSR (Sprague et al., 2024; Suzgun et al., 2023; Hendrycks et al., 2021b;a; Wang et al.,

2024b)) primarily measure question-answering competence. *Tool-augmented evaluations* (e.g., APIBench/Gorilla (Patil et al., 2024), ToolBench (Qin et al., 2024a; Guo et al., 2024), ToolHop (Ye et al., 2025)) instead assess whether LLMs can select and execute appropriate tools, and may further measure end-to-end task completion. *Agent-centric leaderboards* additionally report agents' performance on complex tasks (Lab, 2025; Ma et al., 2025; Bhavsar, 2025).

Despite this wealth of evaluation data, existing efforts treat benchmark results as isolated diagnostic endpoints. Performance metrics are reported per-benchmark with heterogeneous schema annotation formats range from binary correctness flags to numerical scores to execution trace logs. Our work introduces the AGENTSELECT Benchmark, the first unified framework that systematically converts heterogeneous evaluation outcomes into structured preference signals for agent recommendation. This paradigm shift unlocks their prescriptive value—guiding which agent to select for a given task—beyond their traditional diagnostic role.

### 2.2. Existing Evaluation Benchmarks

*LLM QA leaderboards* (e.g., the Open LLM Leaderboard (Beeching et al., 2023; Fourrier et al., 2024), built on suites such as MMLU/BBH/MATH/MUSR (Sprague et al., 2024; Suzgun et al., 2023; Hendrycks et al., 2021b;a; Wang et al., 2024b)) primarily evaluate models' question-answering competence over diverse tasks. *Tool-augmented evaluations* (e.g., APIBench/Gorilla (Patil et al., 2024), ToolBench (Qin et al., 2024a; Guo et al., 2024), ToolHop (Ye et al., 2025)) assess LLMs' ability to solve complex tasks with external tools, including tool selection, argument generation, API execution, and end-to-end task completion. *Agent-centric leaderboards* further extend evaluation to agents operating over complex, multi-step tasks (Lab, 2025; Ma et al., 2025; Bhavsar, 2025).

Despite this wealth of evaluation data, existing efforts primarily use benchmark results as isolated diagnostic endpoints. They report how a model, tool-using system, or submitted agent performs under a fixed benchmark protocol, but do not provide *query-conditioned supervision* for deciding which deployable agent configuration should be selected for a new narrative task. Moreover, their outputs are highly heterogeneous; as a result, these resources are difficult to reuse directly as standardized training data for agent recommendation. In contrast, AGENTSELECT systematically converts heterogeneous evaluation outcomes into unified query–agent interaction signals. This shifts evaluation artifacts from a purely diagnostic role to a prescriptive role: providing reusable supervision for deciding which agent configuration should be recommended for a given task.

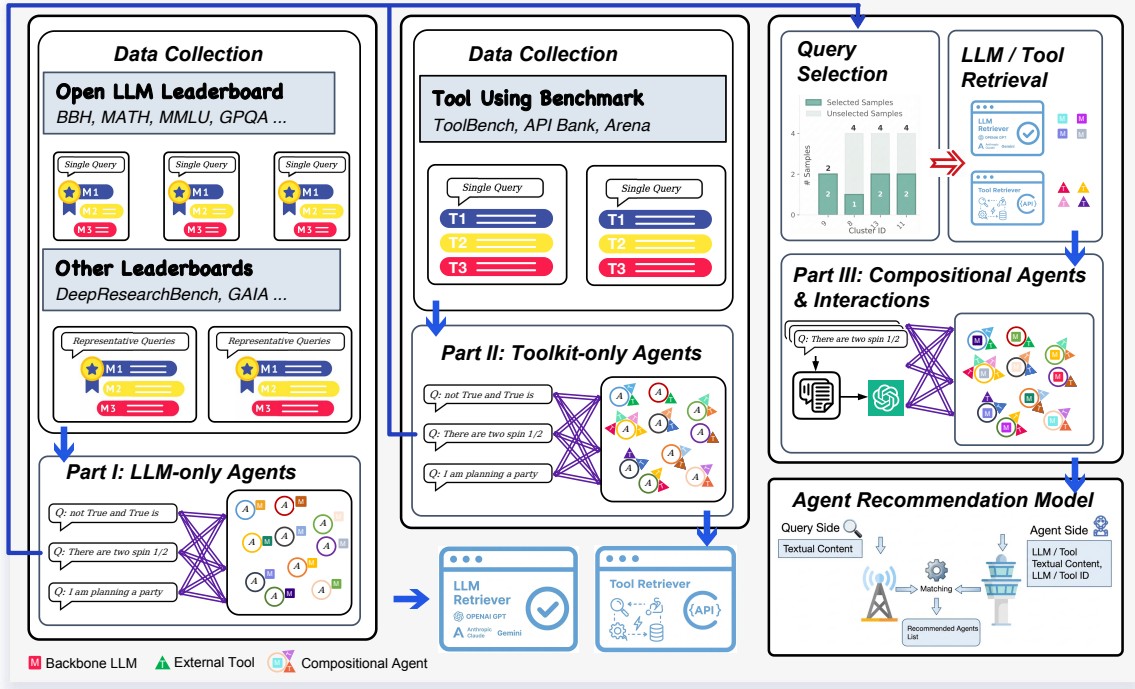

Figure 1. **Overview of AGENTSELECT.** We construct three benchmark parts—LLM-only (Part I), toolkit-only (Part II), and compositional agents (Part III)—and use the resulting interactions to train an agent recommender for natural-language queries. Arrows show the flow; icons indicate backbone LLMs, tools, and composed agents.

## 2.3. Routing and Retrieval for Agent Composition

A growing line of work studies *LLM routing*: given a task, selecting an appropriate model to improve answer quality or optimize quality–cost–latency trade-offs. Recent benchmarks and protocols standardize router evaluation and aggregate large-scale performance evidence, such as RouterBench (Hu et al., 2024) and RouterEval (Huang et al., 2025), alongside advances in optimization-based routing (Mei et al., 2025) and robustness to unseen or newly released models (Jitkrittum et al., 2025). In parallel, as LLMs are increasingly deployed with external tools, another line of work studies *tool retrieval*: retrieving query-relevant tools from large inventories with heterogeneous and often verbose schemas. Representative efforts include ToolRet (Shi et al., 2025), RAG-MCP (Gan & Sun, 2025), MCP-Zero (Fei et al., 2025), and MCPBench (Luo et al., 2025), as well as complementary studies on tool representation learning and tighter retrieval–execution coupling, such as Tool2Vec (Moon et al., 2024) and ToolGen (Wang et al., 2025b).

These two lines address important component-level decisions behind deployable agents: choosing a backbone LLM and selecting a compatible toolset. Together, these components define an agent's capability profile. However, existing routing and tool-retrieval methods mainly narrow the model or tool search space separately; they do not directly provide supervision for ranking complete deployable configurations conditioned on free-form narrative queries. In contrast, we formulate the end-to-end *query-to-agent recommendation* task, where the goal is to select the agent configuration most likely to satisfy a user's task intent. AGENTSELECT supports this formulation by converting heterogeneous evaluation artifacts into structured, positive-only query–agent supervision for learning to recommend agents at scale.

## 3. Task Definition: Narrative Query-to-Agent Recommendation

We take as input a free-form natural-language query $Q$ for retrieving relevant agents. Unlike standard user–item recommendation, we assume no persistent user identity or long-term history; the user's intent, task description, and session context are fully captured in $Q$.

Let $\mathcal{A} = \{A_1, \ldots, A_n\}$ denote the marketplace catalog. The recommender ranks agents in $\mathcal{A}$ by how well they can address $Q$, reflecting the combinatorial and evolving nature of agent inventories: in practice, agents are instantiated by coupling a backbone language model with a set of external tools, yielding a large space of configurations. Formally, the model outputs a top-$k$ ranked list

$$\hat{\mathcal{R}}_k(Q) = \text{Top-}k\big(\{\, s(Q, A) : A \in \mathcal{A}(Q) \,\}\big),$$

where $s(Q, A)$ estimates the utility of agent $A$ for query $Q$, and $\mathcal{A}(Q) \subseteq \mathcal{A}$ is the candidate set used at inference time. The objective is top-$k$ ranking that aligns narrative queries with agent capabilities.

*Table 1.* Example of Agent Configuration.

| Field | Value |
|---|---|
| **Backbone LLM** $M$ | *Name:* `Qwen2.5 Instruct-72B`
*Description:* Released in September 2024, Qwen2.5 pushes the context window to 128k and can generate passages up to 8k tokens. Relative to Qwen2 it shows ... |
| **Toolkit** $T$ | *Name:* `family_relation_finder`
*Description:* A tool designed to find and analyze familial relationships... |
| | *Name:* `genealogy_query`
*Description:* Focusing on identifying various familial connections of historical or contemporary figures. |
| | *Name:* `extract_last_name`
*Description:* Extracting the last name from a full name string... |
| | *Name:* `advanced_character_counter`
*Description:* Counting occurrences of specified characters in given strings... |
| **Configuration** $C$ | *Session History:*
`add_history_to_messages`: True
`read_chat_history`: True
`read_tool_call_history`: True |
| | *Memory Management:*
`enable_agentic_memory`: True
`enable_user_memories`: True
`enable_session_summaries`: True |
| | *Knowledge Integration:*
`knowledge_base`: Some Built Vector Databases |

## 4. AGENTSELECT Benchmark

### 4.1. Capability Profile Design

A central requirement of AGENTSELECT is that each candidate must be a deployable problem-solver rather than an abstract label. We therefore represent an agent by a capability profile $A = (M, T)$, where $M \in \mathcal{M}$ is the backbone language model and $T \subseteq \mathcal{T}$ is the set of external tools the agent can invoke. This abstraction captures the dominant source of functional differences among practical LLM agents: (i) the reasoning and language competence provided by the backbone, and (ii) the actionable interface to the world provided by tools (e.g., search, retrieval, code execution, databases, scheduling, and domain APIs). In our benchmark, recommendation is formulated as *capability matching*, so the profile focuses on $M$ and $T$ as the minimal, directly comparable core.

To bridge a descriptive profile and an operational agent, we store each agent representation as a *YAML configuration file* (see Table 1). The schema explicitly specifies the LLM model, tool combinations, and execution metadata needed for instantiation. This makes each recommended agent actionable: a system can realize the agent by loading the YAML and mapping the declared configurations to a runnable agent in general agent frameworks (e.g., Agno, AutoGen, LangChain, LangGraph).

We store agents as full $(M, T, C)$ YAMLs to make every catalog entry runnable under a standard runtime, since real deployments inevitably require policy-level details such as prompting, memory, decoding parameters, and tool-calling policies. However, $C$ is largely framework- and implementation-specific, often missing or non-portable across sources, and thus cannot be standardized for fair, reproducible comparison. We therefore treat these factors as default or controlled runtime settings, rather than as supervised recommendation targets. Under this design, the benchmark learns and evaluates over the capability core $(M, T)$, which provides the most stable cross-framework abstraction of "what the agent can do".

### 4.2. Dataset Design

We construct a query-to-agent recommendation dataset where each natural-language query $Q$ is paired with a small set of suitable agents $A = (M, T)$ under *positive-only* supervision, mirroring implicit-feedback settings where only observed positives are recorded (Hu et al., 2008; Rendle et al., 2009). Rather than annotating from scratch, we recast heterogeneous community artifacts—LLM leaderboards and tool-use / tool-retrieval corpora—into a unified query–agent interaction format for capability matching over $(M, T)$, so the benchmark can be extended as new artifacts emerge. Concretely, our supervision comes from three complementary signals: (i) model-selection preferences over backbones $M$ extracted from large-scale LLM evaluations when tools are absent, (ii) tool-adequacy evidence over required toolkits $T$ derived from tool benchmarks that specify or imply necessary tools independent of the backbone, and (iii) compositional$(M, T)$ feedback from synthesized pseudo-positive implicit interactions.

**Part I: LLM-only Agents.** *(1) Leveraging Query-Granular Evaluation Results.* We construct LLM-only agents from the Open LLM Leaderboard, which reports per-query evaluations across diverse benchmarks and thus supplies graded scores that function as explicit feedback, akin to ratings in recommender systems. We form query–agent interactions by treating the top-10 ranked agents for each query as positives. To ensure stability, reliability, and practical relevance, we further restrict the full catalog of 4,426 LLMs to models officially released by well-recognized institutions (e.g., Microsoft, NVIDIA, DeepSeek-AI, Google), resulting in a subset of 173 models. *(2) Leveraging Dataset-Granular Evaluation Results.* Many LLM benchmarks report only dataset-level scores (one score per model per dataset), which are too coarse to directly train a query-conditioned agent recommender. We therefore treat these aggregates as task-level priors. For each benchmark dataset, we construct a coverage-balanced coreset of its queries, and assign each coreset query the same per-model ordering implied by the published results. This produces weak but stable supervision that reflects which models are preferred for this task family, while keeping the training data in our unified query–agent interaction format. For the details of sampling process can refer to Appendix E, where we embed queries,

cluster them, allocate adaptive budgets to ensure broad semantic coverage, and select representative prototypes per cluster.

**Part II: Toolkit-only Agents.** Modern LLM agents operate by invoking external tools (functions/APIs), and the choice of tools is often important for solving tool-intensive queries. Accordingly, several recent benchmarks evaluate tool use by either (i) providing an idealized toolkit required to solve each query, or (ii) recording successful tool-use trajectories from which the necessary tools can be inferred. We leverage these existing supervision signals to build backbone LLM-agnostic candidates that isolate the contribution of toolsets. Concretely, for each query, we construct a toolkit-only agent by setting its backbone to a null placeholder and fixing its tool set to the benchmark-provided reference toolkit. The resulting agent serves as the positive target in our recommendation data.

**Part III: Compositional Agents.** Signals can be sourced from backbone–LLM QA results and from corpora with gold toolkits, yet these resources miss practical deployment structure: capabilities are fragmented across modalities $\mathcal{M}$ and $\mathcal{T}$, and direct query–agent interactions over complete $(M, T)$ configurations are scarce. Models trained solely on Part I and Part II therefore struggle to recommend end-to-end agents in realistic settings. Part III addresses this gap by synthesizing compositional agents, i.e., explicit $(M, T)$ configurations, around carefully selected queries and by logging simulated positive interactions that approximate capability-level user choices. Building on these motivations, we generate Part III using a three-stage pipeline.

We select a coverage-balanced set of prototypical queries from Parts I and II and use it as the query set for Part III. For each query, we synthesize an agent-candidate set by first retrieving components and then composing them into $(M, T)$ configurations. We use two lightweight retrievers trained on Parts I/II: an LLM retriever that produces a ranked shortlist of backbone LLMs likely to match the query intent, and a tool retriever that returns a ranked set of tools required to execute the task. For Part I queries, we further decompose the query into multiple facets to surface fine-grained tool needs and aggregate retrieved tools across facets for robustness. Finally, we compose the retrieved backbone and tool shortlists into a query-specific pool of $(M, T)$ agent configurations. The flowchart of the synthesis process is shown in Figure 2.

Part III does not directly annotate every retrieved component or Cartesian-product configuration as a positive label. Instead, retrieved LLMs and tools are first used only as candidate components, and an LLM-assisted synthesis and validation step further composes query-specific configurations that satisfy the inferred capability requirements. We retain these configurations as pseudo-positive interactions.

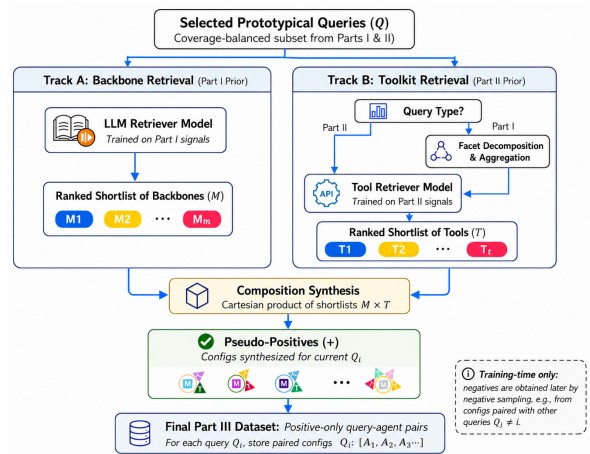

*Figure 2.* Compositional Agent Construction and Query-Agent Interaction Simulation Pipeline for Part III.

Figure 2 provides a visual overview of the overall process. Additionally, we provide human-verification-based reliability analysi in Appendix G.

### 4.3. Benchmark Characteristics

We summarize the key characteristics of our agent recommendation benchmark in Figure 3. Overall, the benchmark contains 111,179 narrative queries and an agent catalog of 107,721 deployable agents (each represented by a capability profile, i.e., backbone model and tool inventory). Supervision is *positive-only*: each query is associated with a small set of suitable agents, yielding 251,103 positive query–agent interactions in total.

**Scale and sparsity across parts.** As shown in Figure 3 a, the benchmark is partitioned into three parts with distinct interaction topologies. Part I contains 23,073 queries but only 231 agents, leading to substantial agent reuse and dense supervision signals per agent. In contrast, Part II and Part III contain 76,197 and 11,909 queries, paired with much larger catalogs of 47,949 and 59,541 agents, respectively. These two parts are therefore markedly sparse over the agent space, and better reflect a realistic long-tail marketplace where many agents are rarely selected. The resulting positive interactions are distributed as: Part I contributes 45.9% of all positives, while Part II and Part III contribute 30.3% and 23.7% (Figure 3a).

**Tool diversity.** The number of unique tools used by agents is 12,099, and for each underlying source, ToolGen (Wang et al., 2025b; Qin et al., 2024a) covers the largest space with 10,939 distinct tools, while other sources provide smaller tool suites (e.g., ToolHop (Ye et al., 2025): 622, UltraTool (Huang et al., 2024): 434, APIBank (Li et al., 2023): 100, Arena (Yekollu et al., 2024): 64).

**Source breakdown within Part I and II.** Figure 3b further breaks down the query composition of Parts I and II. The

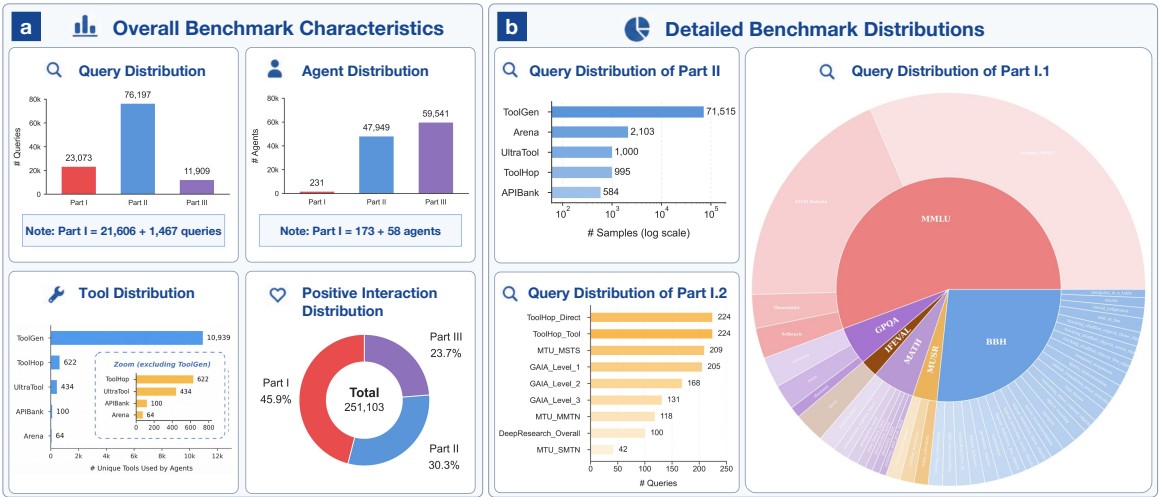

*Figure 3.* Overview of Benchmark Characteristics and Distribution.

benchmark aggregates multiple established sources with substantially different scales (e.g., Arena: 2,103, UltraTool: 1,000, ToolHop: 995, APIBank: 584), with ToolGen being the largest (71,515). For Part I.2, the query distribution is spread across several sub-benchmarks of comparable size such as ToolHop (Ye et al., 2025), MTU (Wang et al., 2024a), GAIA (Mialon et al., 2024), and DeepResearch-Bench (Du et al., 2025). For Part I.1, queries are dominated by MMLU (Hendrycks et al., 2021a) and BBH (Suzgun et al., 2023) (over 75%), with the remainder drawn from GPQA (Rein et al., 2024), IFEVAL (Qin et al., 2024b), MATH (Hendrycks et al., 2021b), and MUSR (Sprague et al., 2024).

## 5. Experimental Setup

In this section, we summarize the representative recommenders evaluated in our benchmark. The compared approaches span six method families, covering a broad range of standard recommendation and retrieval paradigms: (i) interaction-based latent factor models, including MF (Koren et al., 2009) and LightFM (Kula, 2015); (ii) content-aware neural recommendation models (Chen et al., 2024; Lin et al., 2025; Liu et al., 2024; Deng et al., 2025b), including DNN (He et al., 2017) and Two-Tower architectures (Huang et al., 2013); (iii) graph-based propagation methods (Ma et al., 2026), including NGCF (Wang et al., 2019b), KGAT (Wang et al., 2019a), LightGCN (He et al., 2020), and SimGCL (Liu et al., 2023); (iv) embedding-based retrieval and reranking methods, including BGE-Reranking (Chen et al., 2024), KaLM (Zhao et al., 2025), and EasyRec (Ren & Huang, 2024); (v) Agents for Recommendation, e.g., Agent4Rec (Zhang et al., 2024), iAgent (Xu et al., 2025), and Apeer (Jin et al., 2025). In our paper, we adopt a simplified task-adapted variant of this paradigm; and (vi) Generative Recommendation paradigms, such as OneRec (Deng et al., 2025a; Zhou et al., 2025), we simplify the

implementation to directly generate the target agent ID.

All experiments were conducted on NVIDIA A40 GPU using PyTorch 2.2.2 and Transformers 4.48.1. For a fair comparison, all methods are evaluated under a unified feature setting, where query content, model content, tool content, model ID, and tool ID are consistently used. The graph-based method further incorporates agent IDs to construct graph relations. For completeness, we also consider additional variants of MF to TwoTower where agent ID features are explicitly added. For the two factorization baselines and the four graph-based models, we additionally include a query-ID feature; at inference time, we approximate the query-ID embedding by retrieving the $N=3$ nearest training queries using TF-IDF and taking the mean of their query-ID embeddings. The number of positive interactions differs across dataset parts: top-10; Part II: top-1; Part III: top-5. The train-test split ratio is 8:2. We report results separately for Part I/II/III using Precision@10, Recall@10, F1@10, nDCG@10, and MRR@10. Additional implementation details are provided in Appendix F.

## 6. Results and Analysis

We present our main results through (1) the overall leaderboard across Parts I–III (Table 2), (2) the effect of adding agent ID information to non-graph methods (Figure 5), and (3) the long-tailed popularity structure of agents and their reusable content components (Figure 4). From these results we have the following key findings.

**Query/agent IDs help under dense reuse but fail under one-off supervision.** The unnormalized ID-level popularity is sharply head-heavy and is mainly driven by Part I, where positives concentrate on a small set of repeatedly used agents. In contrast, the normalized content-based popularity, computed from reusable LLM and tool components, is much flatter: the top 1%/5%/10% agents explain

*Table 2.* Leaderboard results for query-to-agent recommendation on Parts I–III. Language embedding models marked with * are trained with in-domain supervision (fine-tuned). All reported metrics are calculated based on the top-10 recommendations.

| Method | Part I | | | | | Part II | | | | | Part III | | | | |
|---|---|---|---|---|---|---|---|---|---|---|---|---|---|---|---|
| | Prec. | Rec. | F1 | nDCG | MRR | Prec. | Rec. | F1 | nDCG | MRR | Prec. | Rec. | F1 | nDCG | MRR |
| MF | 0.2937 | 0.2989 | 0.2956 | 0.3094 | 0.5337 | 0.0142 | 0.1417 | 0.0258 | 0.0977 | 0.0843 | 0.0304 | 0.0607 | 0.0405 | 0.0547 | 0.0802 |
| LightFM | 0.3424 | 0.3486 | 0.3447 | 0.3643 | 0.6011 | 0.0697 | 0.6966 | 0.1267 | 0.4776 | 0.4091 | 0.0926 | 0.1852 | 0.1235 | 0.1422 | 0.1610 |
| DNN (TFIDF) | 0.3039 | 0.3099 | 0.3061 | 0.3320 | 0.6123 | 0.0854 | 0.8536 | 0.1552 | 0.6696 | 0.6111 | 0.1808 | 0.3617 | 0.2411 | 0.2975 | 0.3090 |
| DNN (BGEM3) | 0.2970 | 0.3024 | 0.2990 | 0.3165 | 0.5600 | 0.0670 | 0.6701 | 0.1218 | 0.4152 | 0.3364 | 0.1791 | 0.3582 | 0.2388 | 0.2963 | 0.3392 |
| DNN (Bert) | 0.3001 | 0.3079 | 0.3031 | 0.3170 | 0.5399 | 0.0442 | 0.4416 | 0.0803 | 0.2420 | 0.1813 | 0.0564 | 0.1129 | 0.0753 | 0.0818 | 0.0911 |
| DNN (Bert*) | 0.3202 | 0.3279 | 0.3231 | 0.3449 | 0.5868 | 0.0642 | 0.6422 | 0.1168 | 0.4082 | 0.3359 | 0.1169 | 0.2338 | 0.1558 | 0.1797 | 0.1901 |
| TwoTower (TFIDF) | 0.4063 | 0.4149 | 0.4095 | 0.4422 | 0.6791 | 0.0989 | 0.9890 | 0.1798 | 0.9663 | 0.9588 | 0.4268 | 0.8536 | 0.5691 | 0.8427 | 0.9005 |
| TwoTower (BGEM3) | 0.3532 | 0.3618 | 0.3564 | 0.3785 | 0.5971 | 0.0989 | 0.9889 | 0.1798 | 0.9541 | 0.9425 | 0.4197 | 0.8393 | 0.5596 | 0.8268 | 0.8861 |
| NGCF | 0.8777 | 0.8872 | 0.8812 | 0.9045 | 0.9652 | 0.0140 | 0.1398 | 0.0254 | 0.0755 | 0.0559 | 0.0004 | 0.0009 | 0.0006 | 0.0006 | 0.0003 |
| KGAT | 0.8530 | 0.8616 | 0.8562 | 0.8688 | 0.9126 | 0.0240 | 0.2403 | 0.0437 | 0.1721 | 0.1509 | 0.0004 | 0.0009 | 0.0006 | 0.0006 | 0.0003 |
| LightGCN | 0.8642 | 0.8730 | 0.8675 | 0.8820 | 0.9244 | 0.0389 | 0.3886 | 0.0707 | 0.3037 | 0.2765 | 0.0000 | 0.0001 | 0.0001 | 0.0000 | 0.0001 |
| SimGCL | 0.8050 | 0.8137 | 0.8083 | 0.8290 | 0.8868 | 0.0588 | 0.5882 | 0.1069 | 0.4597 | 0.4204 | 0.1011 | 0.2022 | 0.1348 | 0.1376 | 0.1033 |
| BGE-Rerank | 0.0265 | 0.0275 | 0.0269 | 0.0283 | 0.0689 | 0.0530 | 0.5300 | 0.0964 | 0.4260 | 0.3932 | 0.1775 | 0.3550 | 0.2367 | 0.3163 | 0.3307 |
| BGE-Rerank* | 0.1370 | 0.1373 | 0.1371 | 0.1468 | 0.3560 | 0.0920 | 0.9200 | 0.1673 | 0.7800 | 0.7349 | 0.2930 | 0.5860 | 0.3907 | 0.5407 | 0.5658 |
| KaLM-v2.5 | 0.0170 | 0.0170 | 0.0170 | 0.0164 | 0.0321 | 0.0890 | 0.8900 | 0.1618 | 0.8129 | 0.7881 | 0.3375 | 0.6750 | 0.4500 | 0.6189 | 0.6269 |
| KaLM-v2.5* | 0.2850 | 0.2950 | 0.2888 | 0.2787 | 0.4052 | 0.0970 | 0.9700 | 0.1764 | 0.8859 | 0.8584 | 0.4255 | 0.8510 | 0.5673 | 0.8116 | 0.8352 |
| EasyRec | 0.0150 | 0.0150 | 0.0150 | 0.0155 | 0.0353 | 0.0550 | 0.5500 | 0.1000 | 0.3494 | 0.2862 | 0.1865 | 0.3730 | 0.2487 | 0.3115 | 0.3105 |
| EasyRec* | 0.2565 | 0.2632 | 0.2590 | 0.2708 | 0.4969 | 0.0965 | 0.9650 | 0.1755 | 0.8552 | 0.8193 | 0.3405 | 0.6810 | 0.4540 | 0.6320 | 0.6501 |
| AgentRec (KaLM-v2.5*) | 0.2820 | 0.2920 | 0.2858 | 0.2732 | 0.3805 | 0.0970 | 0.9700 | 0.1764 | 0.8859 | 0.8584 | 0.4260 | 0.8520 | 0.5680 | 0.8118 | 0.8353 |
| GenRec | 0.9215 | 0.9255 | 0.9230 | 0.9404 | 0.9925 | 0.0720 | 0.7200 | 0.1309 | 0.5407 | 0.4849 | 0.1585 | 0.3170 | 0.2113 | 0.2780 | 0.3509 |

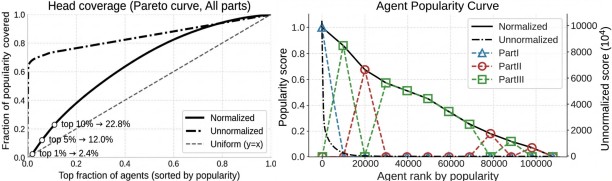

*Figure 4.* Long-tailed agent popularity in our benchmark. **Left:** Pareto curve over all parts. **Right:** popularity-by-rank curves, with Part I/II/III overlaid. We report two definitions: *Unnormalized* popularity counts repeated occurrences of the same agent ID, while *Normalized* popularity is computed from content side – the reusable agent components, and scaled to $[0, 1]$.

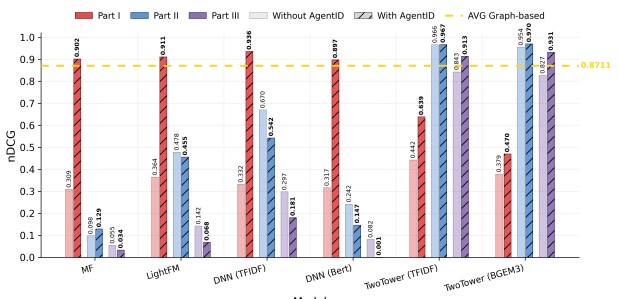

*Figure 5.* Comparison of nDCG performance w/o AgentID.

only 2.4%/12.0%/22.8% of the content-based popularity mass. This suggests that Part I contains strong collaborative-filtering signals arising from dense query–agent interactions, whereas Parts II/III largely fall into a one-off recommendation regime. This distributional shift is consistent with the experimental results in Table 2 and Figure 5. Graph-based methods and AgentID-enhanced variants perform particularly well on Part I, while adding AgentID often hurts MF, LightFM, and DNN on Parts II/III because many validation agents are unseen, making identity embeddings vulnera-

ble to cold-start. TwoTower is the main exception. Since all models use the same features and data, this difference is more likely due to how the signals are fused and optimized: TwoTower learns query and agent representations in a shared retrieval space, where AgentID can help separate agents with similar LLM/tool/content representations rather than merely memorizing popularity. This may explain why AgentID remains useful for TwoTower on Parts II/III. A similar pattern is observed for query-ID based CF/Graph methods: they are competitive when dense query–agent co-occurrence exists in Part I, but become fragile on Parts II/III, where unseen queries and rarely repeated agents collapse shared-neighborhood signals and make query-ID compression ill-suited to near one-off supervision.

**For neural content-based models, explicit TF-IDF features are already highly competitive.** Content-aware DNN and Two-Tower models remain effective on Parts II/III by directly matching narrative user intents with explicit agent capability profiles. We also find that within this neural content-based setting, stronger pretrained encoders do not lead to monotonic gains. In particular, TwoTower (TF-IDF) achieves the best overall performance across all three parts, including Part III, while TwoTower (BGE-M3) remains competitive but slightly lower. Even end-to-end trained DNN (BERT*) is less stable. This suggests that agent capability descriptions often contain highly discriminative lexical cues, such as model names, tool names, and functional terms; therefore, sparse TF-IDF features provide a strong and efficient explicit representation. Dense features are not always more effective than explicit sparse lexical features in the content-aware neural recommendation setting,

**For embedding-based matching, stronger and tuned en-**

**coders improve long-tail alignment but remain limited under ID-driven reuse.** Stronger text representations provide clear benefits for embedding-based matching. We observe a consistent trend from BGE-Rerank to EasyRec and KaLM-v2.5, with larger gains on Parts II/III, where recommendation depends more on matching free-form narrative intents to fine-grained agent capabilities. This trend becomes more evident after in-domain tuning, which helps close the gap by aligning the query semantic space with benchmark-specific configuration distinctions. However, both before and after tuning, embedding-based methods remain limited on Part I.

**Agentic recommendation: limited gains for agent selection.** Building AgentRec on top of KaLM-v2.5$^*$ yields almost no additional improvement. This is expected for two main reasons. (1) First, prior agentic recommender designs are largely user-centric and therefore do not align well with our setting. For example, iAgent mainly improves instruction following, memory, and self-reflection on the user side, while Agent4Rec focuses on modeling long-horizon user behavior through profile, memory, and action modules. By contrast, our task is a single-shot narrative query-to-agent matching problem, where the main challenge is fine-grained discrimination among candidate agent configurations rather than modeling evolving user preferences through interaction. (2) Second, although agentic reranking may improve recommendation quality in conventional domains such as shopping or music, this advantage does not transfer naturally to the fundamentally different setting where agents themselves are the items. A key reason is that agent selection remains a setting with little supervision available, so existing LLMs failed to capture capability distinctions required here.

**Generative recommendation: benefits from reuse; biased under sparse positives.** OneRec is strong on Part I but weaker on Parts II/III, where sparse positives over a large catalog encourage over-selection of frequently observed IDs (cf. the long-tail structure in Figure 4). Our OOD diagnostics reflect this tendency: 0.80/0.71 of predicted top-$K$ items in Parts II/III are training-seen agents, although unseen agents are still occasionally selected. This highlights a limitation of direct agent-token generation under one-off supervision; generating an intermediate capability profile followed by retrieval/re-ranking may be a more suitable direction, which we leave as future work.

## 7. Modality Attribution: IDs vs. Text

Our recommender combines query text with agent-side content features and identity features. The content features describe the capabilities of the backbone model and tools, whereas the identity features encode their discrete IDs. This setting raises a key attribution question: whether ranking accuracy comes from matching user intents to agent-side content features alone, or from exploiting reusable ID-level regularities. To isolate these effects, we conduct controlled feature ablations that remove model/tool content and model/tool IDs under the same evaluation protocol.

From Table 3, we observe that the full feature set achieves the strongest performance, with 0.9344 nDCG and 0.9453 MRR on Part III. Removing the LLM ID, tool ID, or both IDs substantially reduces nDCG to 0.6162, 0.5903, and 0.6071, respectively, indicating that textual capability descriptions alone cannot recover the full ranking signal. Conversely, IDs alone are also insufficient: the IDs-only setting reaches only 0.5744 nDCG and 0.6300 MRR, well below the full model. These results show that strong ranking performance depends on the complementarity between textual capability matching and reusable identity signals.

The two ID modalities further exhibit a clear asymmetry. Tool ID only achieves 0.6040 nDCG and 0.6607 MRR, whereas LLM ID only obtains 0.0857 nDCG and 0.1612 MRR, suggesting that tool identity carries substantially stronger task-specific information than backbone identity in Part III. Overall, the results cannot be explained by either content-only matching or ID-only memorization; effective agent recommendation requires jointly modeling textual descriptions and structured model/tool identities, with tool-related identity providing the dominant discrete signal.

## 8. Effectiveness Validation for Pseudo-Positive Interactions

This section validates that Part III pseudo-positives provide reliable supervision. Part III interactions can be viewed as positive-only implicit feedback over $(M, T)$ configurations. We present behavioral evidence through counterfactual capability edits: the learned ranker shifts its preferences in the expected directions, and a model trained on Part III is both accurate and sensitive to subtle configuration changes, indirectly supporting the quality of the generated supervision. Finally, we show that Part III contributes additional learnable structure beyond Parts I/II, improving coverage over realistic compositions. The experimental results are reported in Table 4 and Table 5, with the corresponding analysis provided in Appendix A.

## 9. Practical Real-World Validation

### 9.1. Validation on MuleRun Agent Marketplace

We further evaluate whether supervision learned from AGENTSELECT transfers to a realistic external agent marketplace. Specifically, we compare EasyRec$^*$, fine-tuned on AGENTSELECT, against the original EasyRec on MuleRun, an unseen public marketplace containing 100+ task-oriented agents. This experiment is not intended as a transfer-based

*Table 3.* Modality attribution via feature ablations (Top@10). Query text is always included; we ablate agent-side features. "LLM/Tool content" refers to the textual descriptions used to represent the backbone model and toolkit. "LLM/Tool ID" are discrete identifiers. All results are obtained with the TwoTower TF-IDF model trained on Part III and evaluated on the held-out Part III split.

| | Features | | | | Validation (Part III / Overall) | | | | | |
|---|---|---|---|---|---|---|---|---|---|---|
| Setting | LLM content | Tool content | LLM ID | Tool ID | P | R | F1 | Hit | nDCG | MRR |
| All modalities (content + IDs) | ✓ | ✓ | ✓ | ✓ | 0.4753 | 0.9506 | 0.6337 | 0.9830 | 0.9344 | 0.9453 |
| No LLM ID | ✓ | ✓ | ✗ | ✓ | 0.3244 | 0.6488 | 0.4325 | 0.8354 | 0.6162 | 0.6709 |
| No Tool ID | ✓ | ✓ | ✓ | ✗ | 0.3182 | 0.6365 | 0.4243 | 0.8333 | 0.5903 | 0.6358 |
| No IDs (LLM ID + Tool ID removed) | ✓ | ✓ | ✗ | ✗ | 0.3238 | 0.6476 | 0.4318 | 0.8408 | 0.6071 | 0.6565 |
| IDs only (no content) | ✗ | ✗ | ✓ | ✓ | 0.3052 | 0.6104 | 0.4069 | 0.8135 | 0.5744 | 0.6300 |
| Tool ID only | ✗ | ✗ | ✗ | ✓ | 0.3092 | 0.6183 | 0.4122 | 0.7715 | 0.6040 | 0.6607 |
| LLM ID only | ✗ | ✗ | ✓ | ✗ | 0.0429 | 0.0858 | 0.0572 | 0.2604 | 0.0857 | 0.1612 |

*Table 4.* Counterfactual capability sensitivity on $N{=}100$ queries. Starting from $A_{\text{full}}$, we apply a single controlled intervention to form $A_{\text{cf}}$. A capability-sensitive ranker should show positive score/rank drops and high consistency. We aggregate three metrics: score drop $\Delta s$, rank degradation $\Delta r$, and consistency $\mathbb{I}[s(Q, A_{\text{full}}) > s(Q, A_{\text{cf}})]$

| Intervention ($A_{\text{cf}}$) | $\mathbb{E}[\Delta s]$ ↑ | $\mathbb{E}[\Delta r]$ ↑ | Consistency ↑ |
|---|---|---|---|
| Remove key tool | $0.0538 \pm 0.0335^*$ | $1.76 \pm 0.84^*$ | $64.0 \pm 16.7\%$ |
| Remove secondary tool | $0.0602 \pm 0.0186^{**}$ | $1.72 \pm 0.77^*$ | $64.0 \pm 16.7\%$ |
| Add irrelevant tool | $0.0825 \pm 0.0203^{**}$ | $2.48 \pm 0.67^{**}$ | $84.0 \pm 16.7\%^*$ |
| Add redundant tool | $-0.0287 \pm 0.0387$ | $-0.08 \pm 0.92$ | $44.0 \pm 26.1\%$ |
| Swap backbone (rank 2–5) | $0.0235 \pm 0.0043^{**}$ | $1.28 \pm 0.58^*$ | $92.0 \pm 11.0\%^{**}$ |

Mean±std over 5 runs. $^*$/$^{**}$ denote Holm-corrected $p < 0.05/0.01$.
One-sample $t$-test vs. 0 for $\Delta s, \Delta r$; vs. 50% for Consistency.

*Table 5.* Learnability attribution of Part III with TwoTower (TF-IDF). We train on Parts I+II, Parts I+II+III, and Part III, and evaluate on the held-out splits of Parts I/II/III. $\Delta$ (%) denotes the relative change (each row is compared against the row above).

| Training Data | Eval on Part I | | Eval on Part II | | Eval on Part III | |
|---|---|---|---|---|---|---|
| | nDCG@10 | $\Delta$ % | nDCG@10 | $\Delta$ % | nDCG@10 | $\Delta$ % |
| Train I+II | 0.3399 | – | 0.9771 | – | 0.1329 | – |
| Train I+II+III | 0.4422 | +30.10 | 0.9663 | -01.11 | 0.8427 | +534.09 |
| Train III | 0.1100 | -75.12 | 0.3263 | -66.23 | 0.8267 | -1.90 |

*Table 6.* Real-world transfer on MuleRun: EasyRec$^*$ (tuned on AGENTSELECT) vs. original EasyRec across different cutoffs.

| Cutoff | Method | P | R | F1 | nDCG | MRR |
|---|---|---|---|---|---|---|
| @1 | EasyRec | 0.6259 | 0.6259 | 0.6259 | 0.6259 | 0.6259 |
| | EasyRec$^*$ | **0.7389** | **0.7389** | **0.7389** | **0.7389** | **0.7389** |
| @5 | EasyRec | 0.1630 | 0.8148 | 0.2716 | 0.7297 | 0.7010 |
| | EasyRec$^*$ | **0.1746** | **0.8731** | **0.2910** | **0.8147** | **0.7949** |
| @10 | EasyRec | 0.0866 | 0.8657 | 0.1574 | 0.7462 | 0.7079 |
| | EasyRec$^*$ | **0.0894** | **0.8944** | **0.1626** | **0.8216** | **0.7977** |

validation of whether AGENTSELECT learns capability-matching signals that remain useful beyond the constructed training environment.

To build this evaluation, we sample a subset of MuleRun agents and align them to the Toolkit-only setting in Part II by inferring 5–10 tool primitives from each agent description while fixing the backbone LLM to GPT-5. For each target agent, we then construct a set of natural-language requests that simulate realistic user needs with controlled diversity in phrasing, difficulty, and distractor noise, so that retrieval requires fine-grained capability discrimination rather than simple keyword matching.

As shown in Table 6, EasyRec$^*$ consistently outperforms the original EasyRec, which is designed for general recommendation scenarios, across different cutoff values, yielding better top-$k$ retrieval performance and stronger overall ranking quality on this unseen agent catalog. These results suggest that AGENTSELECT captures transferable supervision rather than merely fitting benchmark-specific artifacts, enabling more effective matching between open-ended user requests and agent capabilities in realistic marketplace settings.

**9.2. Validation on Deployed Agents on Agno**

Complementing this marketplace transfer study, we additionally validate whether the recommender's ranking aligns with actual end-to-end task performance after deploying the

recommended agents, as reported in Appendix B.

## 10. Conclusion

In this work we define the *narrative query-to-agent recommendation* task: selecting a deployable agent configuration for a free-form user request. We introduced AGENTSELECT, a unified benchmark that standardizes heterogeneous evaluation artifacts into query-conditioned, positive-only supervision over capability profiles $(M, T)$, spanning LLM-only, toolkit-only, and compositional agents at scale. Our analyses reveal a regime shift from dense head reuse to long-tail, near one-off supervision, under which content-aware capability matching is essential and Part III provides complementary, learnable signal beyond Parts I/II. We further validated Part III via counterfactual capability edits and showed practical transfer to a real agent marketplace (MuleRun), improving retrieval quality on an unseen catalog. Overall, AGENTSELECT offers a foundation for training and evaluating agent rankers, routers, and tool retrievers, and provides a reusable infrastructure to support the emerging agent ecosystem.

## Acknowledgements

This work was sponsored by the `Australian Research Council under the Linkage Projects Grant LP210100129`.

## Impact Statement

This work supports a plausible next step for agent ecosystems: agent marketplaces and applications that can reliably surface high-quality, deployable agent configurations for a user's narrative request. By benchmarking query-to-agent recommendation and releasing structured resources for training and evaluation, we aim to reduce the expertise required to assemble customized agents, moving from expert-driven composition toward more accessible, on-demand task solutions for everyday users.

Broader impacts depend on deployment. Better agent selection can increase productivity and lower barriers to tool-augmented assistance, but it may also amplify misuse if powerful tools are exposed without appropriate safeguards. Our contribution is an offline evaluation and benchmarking resource built from public artifacts, under fixed capability profiles and static candidate pools. Responsible real-world systems should therefore pair such recommenders with tool permissioning, monitoring, and policy controls.

Looking forward, this benchmark also opens several directions beyond static query-to-agent ranking. Future work can extend the setting toward dynamic environments where available models, tools, user needs, and deployment constraints evolve over time. Another important direction is to model the bundle effect among components (Tu et al., 2026; Ma et al., 2024; Deng et al., 2026), i.e., compositional bundles that jointly consider model choice and tool selection. Finally, generative recommendation (Huang et al., 2026; Gao et al., 2026; Wang et al., 2025a) represents a highly promising direction, where the system moves beyond ranking candidates from a fixed pool and instead composes suitable agents from open agent resources according to task-specific requirements.

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

## Limitations and Future Work

AGENTSELECT focuses on the selection layer of agent operation systems: given a narrative request, it learns to rank deployable capability profiles $(M, T)$ so that appropriate configurations surface early. Accordingly, our evaluation is intentionally aligned with information-retrieval practice (e.g., nDCG/MRR/Hit), which directly measures whether the benchmark supports reliable capability matching and recommendation. In this scope, the benchmark does not aim to exhaustively validate end-to-end task completion by executing every recommended agent configuration. This choice reflects a practical reality of today's agent ecosystem: tool runtimes and external APIs are heterogeneous, rapidly changing, and often difficult to reproduce at benchmark scale, making fully standardized execution a moving target rather than a stable basis for long-term comparison.

Looking forward, a compelling extension is to incorporate selective end-to-end feedback as an additional supervision channel, rather than a replacement. Execution outcomes and traces can supply precisely the kinds of hard learning signals that text-only capability descriptions may miss: near-tie agent pairs that appear similarly plausible under static descriptions but diverge in tool-call correctness, robustness, or success rate in practice. Integrating such signals would enable constructing more discriminative training data (e.g., hard negatives or near-tie preferences), helping the ranker separate subtle configuration differences and improving recommendation reliability in the most ambiguous cases.

Finally, our open-source repository is under active refinement. We plan to provide a lightweight web entry point where users can input a narrative query and receive a recommended agent configuration (backbone model and toolset); the demo link and instructions will be accessible from the public codebase as it is finalized.

## Data Usage Statement

All data used in this work are derived from publicly available benchmarks, leaderboards, and tool-use corpora released by their respective authors and institutions. We access and process these resources solely for academic research and reproducibility, and we do not use any proprietary, private, or commercially licensed datasets. We follow the original terms of use and licenses for each source; when redistribution of upstream content is restricted, we release only derived annotations/statistics and provide references or scripts that enable others to obtain the raw data directly from the original providers.

Our benchmark construction relies on task/query texts and model/tool metadata that are already publicly disclosed in the upstream artifacts. We do not intentionally collect personal or sensitive user information, and the dataset is not designed for identifying individuals. All trademarks, dataset names, and model/tool names remain the property of their respective owners.

## AI Usage Statement

We used GPT to assist with English polishing and phrasing revisions of the manuscript. We also used Nano Banana to help draft the workflow/overview diagram; all technical content, claims, and final figures were verified and finalized by the authors.

## Appendices

## A. Effectiveness Validation for Pseudo-Positive Interactions Details

### A.1. Part III as Positive-Only Implicit Feedback

Unlike Parts I/II, Part III does not execute agents to globally rank the full combinatorial space. Instead, it retrieves a small set of relevant components (backbone models and tools), composes candidate $(M, T)$ configurations, and uses an LLM judge to validate plausibility. Under this retrieval-constrained pipeline, a pseudo-positive indicates a configuration that plausibly satisfies the query's inferred capability requirements, not a certified globally optimal choice. Establishing global optimality would require large-scale, reliable end-to-end execution across heterogeneous APIs and task-specific success criteria, which is operationally difficult and orthogonal to our benchmark goal. This interpretation matches implicit-feedback recommendation: observed interactions provide positive evidence, while unobserved items are unlabeled.

### A.2. Counterfactual Sensitivity as Behavioral Validation

Since execution-based validation is impractical at benchmark scale, we probe capability awareness through controlled counterfactual edits. Starting from a capability-complete agent $A_{\text{full}}$, we apply one intervention to form $A_{\text{cf}}$ and test whether the ranker prefers $A_{\text{full}}$.

As shown in Table 4, capability-reducing edits lead to directionally correct degradations: removing a key or secondary tool reduces scores and worsens ranks, and swapping the backbone to a lower-ranked candidate degrades performance with high consistency (92%). Adding an irrelevant tool is penalized most strongly and consistently (84%), reflecting sensitivity to tool–intent mismatch. In contrast, adding a redundant tool has a near-zero effect (44% consistency with a slightly negative mean $\Delta s$), which is desirable because true redundancy should not systematically change capability. Together, these counterfactual results provide practical evidence that models trained on our benchmark learn capability-aware preferences over subtle configuration differences.

### A.3. Learnability and Non-redundancy of Part III Supervision.

We test whether Part III pseudo-positives provide meaningful supervision by measuring learnability and cross-part transfer. Table 5 shows that Part III is highly learnable: training on Part III alone performs best on Part III, and adding Part III to I+II yields a large gain on Part III. This behavior is inconsistent with a noise-dominated signal and indicates coherent, task-aligned structure.

Notably, mixing Part III with I+II changes performance on Part I. We do not interpret this as "wrong" labels; rather, Part III introduces a different supervision regime centered on compositional capability matching from retrieved components. Under a shared model, this can shift some Part I queries toward plausible Part III-style candidates, which may be penalized by Part I's ground-truth protocol. Consistently, Part III-only training transfers poorly to Parts I/II, confirming that Part III is non-redundant and captures complementary signal rather than restating Parts I/II. Overall, Part III expands supervision coverage over realistic $(M, T)$ configurations while remaining learnable and structurally distinct.

## B. Validating Recommendation Rankings via End-to-End Deployed Agent Performance

This appendix provides a deployment-centric validation. While §9.1 evaluates recommendation quality from an agent-platform perspective, here we test whether the recommender's ordering remains meaningful after the recommended agent configurations are instantiated and executed end-to-end.

We begin with the ranked outputs of the TwoTower (TF-IDF) recommender and execute a realistic shortlist of candidates. Concretely, we sample 200 queries from Part III. For each query, TwoTower produces a ranked list of agent configurations (sorted by predicted utility score), from which we select ranks $\{1, 4, 7, 10, 13\}$ (1-indexed) as the executed top-5 set. Each selected candidate is then deplyoyed into a runnable agent using Agno by injecting its backbone LLM and toolset into the YAML template. We launch the resulting agent as a server process and send the full user request through the runtime API, recording the whole process responses.

To make tool execution repeatable while preserving realistic tool-call structure, we connect the runtime to MIRRORAPI (Guo et al., 2024), which simulates API responses via specialized LLM "mirrors" of tool environments. End-to-end performance is assessed at the task-result level: for each query, we rank the five executed outputs by overall task completion and tool executability under a hybrid protocol (human verification assisted by an LLM judge driven by GPT-5.2). We treat this per-query ordering as the end-to-end label (E2E-LABEL) and evaluate how well the recommender's predicted ordering over the same executed candidates aligns with it. Because runtime execution can be stochastic, we repeat the entire evaluation three times and report averages; we further restrict to queries for which all five candidates yield valid, comparable outcomes.

Table 7 summarizes rank alignment between the recommender ordering (REC) and E2E-LABEL, together with a random-permutation baseline over the same candidate set. Importantly, this is a deliberately challenging regime: the executed candidates are already "high potential" by construction because they come from the top of the recommender's own list, and thus tend to be plausible and competitive. Fine-grained separation within such a strong shortlist is inherently difficult; correspondingly, even random permutations can achieve relatively high nDCG in a top-5 setting, and large absolute gains should not be expected. Despite this, REC yields consistent improvements over random, with the clearest lift concentrated at the very top of the ranking—the regime that matters most in deployment when the system selects a single configuration to run.

| Metric | REC vs E2E-LABEL | Random vs E2E-LABEL | Gain over Random |
|---|---|---|---|
| Top-1 match ↑ | 0.364 [0.236, 0.491] | 0.236 [0.127, 0.345] | +53.9% |
| Spearman $\rho$ ↑ | 0.225 [0.111, 0.336] | −0.011 [−0.140, 0.109] | +0.236 |
| Kendall $\tau$ ↑ | 0.185 [0.095, 0.276] | −0.007 [−0.109, 0.087] | +0.193 |
| nDCG@1 ↑ | 0.695 [0.632, 0.759] | 0.608 [0.548, 0.672] | +14.2% |
| nDCG@3 ↑ | 0.819 [0.787, 0.850] | 0.765 [0.731, 0.798] | +7.1% |
| nDCG@5 ↑ | 0.907 [0.890, 0.923] | 0.880 [0.864, 0.897] | +3.0% |

*Table 7.* End-to-end rank alignment on deployed agents (Agno + MIRRORAPI). We compare the recommender's predicted ordering against the end-to-end judged ordering over executed top-5 candidates, and include a random-permutation baseline over the same set. Values are mean with bootstrap 95% CI; we retain only queries where all five candidates yield valid, comparable outcomes. "Gain over Random" is reported as relative improvement for Top-1 and nDCG (%): $100 \times (\text{REC} − \text{RAND})/\text{RAND}$; for correlation metrics we report absolute differences since the random baseline is near zero and percentage gains are numerically unstable.

Overall, these results indicate that the recommender's ranking signal meaningfully transfers to a deployed setting with tool execution and runtime constraints. The positive correlations with E2E-LABEL and the consistent improvements at the top of the list suggest that the recommender is not only producing a reasonable shortlist, but is also ordering competitive candidates in a way that increases the likelihood of end-to-end task success. Through this validation, people can move beyond manual LLM/tool toggles toward an adaptive system (e.g.,ChatBox) that composes an effective agent configuration on demand for each narrative query.

## C. Practical Significance and Real-World Validation Details

We further evaluate whether a recommender trained on AGENTSELECT generalizes to a real-world, out-of-domain marketplace. We conduct an external case study on MuleRun[2], a public agent marketplace with 100+ task-oriented agents. We sample a subset of publicly listed agents and map them into the Toolkit-only schema of Part II: each agent is represented by 5–10 tool primitives inferred from its description, while fixing the backbone LLM to GPT-5. We author 20 natural-language requests that each unambiguously targets a specific agent niche (e.g., *Mecha Pet*: "I need a cyberpunk-themed portrait of my dog for Instagram."), and evaluate retrieval success by whether the intended agent appears in the top-$k$ list, reporting Top@1/5/10 and ranking metrics (nDCG/MRR) in Table 6. EasyRec* (fine-tuned on AGENTSELECT) consistently outperforms the untuned EasyRec, improving both hit-oriented and ranking metrics (e.g., Precision@1 from 0.3556 to 0.4000, with nDCG/MRR gains across all $k$). Despite the small scale and the use of tool primitives inferred from marketplace text, the consistent improvements on an unseen catalog provide evidence that AGENTSELECT yields transferable supervision and can improve practical agent retrieval in a realistic marketplace setting.

## D. Online Demonstration of the Recommended Agent Configuration

To providean intuitive view of how our recommendations can be used in practice, we build a lightweight online demo that turns the top-ranked result into a runnable agent ( Figure 6). Given a user request, the system outputs a recommended backbone–tool configuration $(M, T)$ and then materializes it into a concrete runtime policy $C$, which specifies execution-time details. We provide glue code in our doe repository that maps the abstract agent configuration into an executable agent endpoint. To ensure that the recommended tools are executable in practice, we suggest invoking an LLM-based simulated tool server (e.g., StableToolBench-MirrorAPI (Guo et al., 2024)) to emulate tool responses, since fully deploying the entire tool suite and guaranteeing end-to-end executability is a substantial engineering effort.

## E. Coverage-Balanced Prototypical Queries Selection

To approximate query-level supervision from dataset-level aggregates, we transform query–aggregate pairs into pseudo query–individual pairs through a targeted data-selection pipeline. The central challenge is to sample benchmark queries that remain faithful to the aggregate scores while spanning the semantic space. Prototype-based selection, effective in vision (Sorscher et al., 2022), tends to concentrate near cluster "centers," yielding homogenized sets that erode diversity and can induce a gap between the benchmark's true mean and the proxy mean measured on the sampled subset. Coverage-centric stratified sampling (Zheng et al., 2023) instead prioritizes comprehensive coreset coverage. We combine these ideas: queries

---

[2]https://mulerun.com/

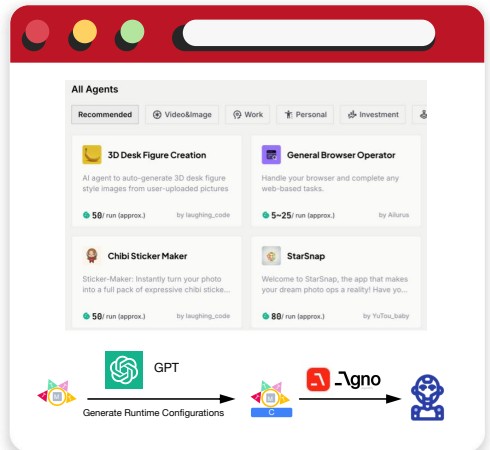

Figure 6. Online deployment of the recommended agent configuration, including the final runtime policy $C$ finalize and glue code for Agno integration.

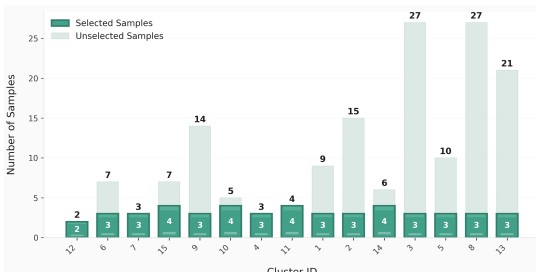

Figure 7. Coverage-Balanced Prototype-Based Data Selection Distribution Selected VS Total Samples.

are embedded with KaLM-v2.5, clustered hierarchically, and each cluster receives an adaptive sampling budget following (Zheng et al., 2023); within clusters, we select prototypes in the spirit of (Sorscher et al., 2022). This yields balanced diversity with global-distribution fidelity and highlighted local patterns, via adaptive per-stratum budgets and prototype selection that maximizes coverage yet stays aligned with the aggregates. Figure 7 sketches the procedure; each bar represents clusters and highlighted segments indicate chosen prototypes.

Importantly, coreset selection is not meant to infer fine-grained per-query outcomes; it is used to expose diverse query surface forms for the same task prior, which improves generalization at inference time. We embed queries, cluster them, allocate adaptive budgets to ensure broad semantic coverage, and select representative prototypes per cluster. Across benchmarks, these task-level priors differ and thus provide discriminative supervision; the recommender learns from query text to associate inputs with the most compatible capability regime and the corresponding model preference.

## F. Evaluation Protocol & Implementation Details

**Universal Setting.** To minimize confounding factors across methods, we unify the input signals and evaluation pipeline. Unless otherwise noted, every recommender is trained and evaluated using the same core inputs: (i) the natural-language query, (ii) the backbone LLM text, (iii) the Tool text, and (iv) the corresponding model/tool identifiers. Tool text is formed by concatenating each tool's name and description, and model text is the LLM name string. In addition, the two factorization baselines and the four graph-based recommenders exploit a learned query-ID embedding when the dataset provides query identifiers. We do not use agent ID as an explicit input feature for model training. The only exception is GenRec, where each agent is mapped to a discrete token in the generative output space. Query identifiers are unavailable for truly unseen queries, so at inference time we approximate the query-ID embedding with a TF–IDF nearest-neighbor surrogate. Specifically, for each test query we retrieve its $N=3$ most similar training queries under TF–IDF cosine similarity, and use the similarity-weighted average of their learned query-ID embeddings as the test-time representation. This surrogate is only applied to models that require query IDs; all other methods directly consume the raw query text.

**Specific Implementation.** For text-encoding baselines based on BERT, we use DistilBERT as the encoder backbone. For GenRec, we implement a simplified variant consistent with the paper's high-level formulation but without the semantic-ID generation and subsequent matching stage. Concretely, we discretize each agent as a vocabulary token, score a retrieved candidate pool using candidate-only logits, and output Top-10 recommendations in a single forward pass. Training uses a multi-positive one-step softmax objective, augmented with an in-batch multi-positive InfoNCE term to sharpen the representation space. Starting from the SFT checkpoint, we further apply DPO, where preference pairs are constructed from a ground-truth-based reward derived from nDCG@K. Finally, we warm-start the agent-token embedding table with representations learned from a Two-Tower BGE model, providing a semantically informed initialization that improves convergence under limited supervision. We implement AgentRec by adapting the agentic recommendation paradigms of iAgent and Agent4Rec to our narrative-driven setting. We first use KaLM-v2.5* to coarsely rank a pool of 1,000 candidate agents, and then rerank the top 30 candidates with an agentic model based on GPT-5.4-nano. This design avoids the impractical cost of directly applying agentic reasoning to all 1,000 candidates, while remaining effective because KaLM-v2.5 already achieves strong top-rank recall. Owing to the high inference cost of agentic recommendation, we evaluate AgentRec on 1,000 sampled instances rather than the full benchmark, which is already substantial for this class of methods. Different from Agent4Rec, we do not explicitly model page-by-page browsing, as the long-context model can directly process the reranking candidates in a single pass. In addition, separate profile, intent, and memory modules are not explicitly introduced, since in our narrative-driven task such information is largely expressed in the query itself, including task intent and session history.

**Evaluation.** We follow the dataset supervision design and keep the number of positives per query fixed by part: top-10 positives for Part I, top-1 for Part II, and top-5 for Part III. We evaluate Part I/II/III separately and report Precision@10, Recall@10, F1@10, nDCG@10, and MRR@10. To ensure consistent comparison under large candidate catalogs, we adopt sampled evaluation and fix the candidate pool size to 1,000 for all methods.

## G. Reliability Analysis for Part III Construction

Since the pseudo-positive compositional agents in Part III are constructed by retrieving candidate components and then synthesizing them with the assistance of an LLM, a natural concern is whether the resulting supervision may inherit bias from this pipeline. To examine this issue, we conduct a targeted human verification study on a sampled subset of Part III instances. The goal here is not to argue that the synthesized labels are fully noise-free, but to understand whether they are reliable enough for the benchmark's intended use and how far they may deviate from human judgment.

Our evaluation includes two subsets of Part III: 195 sampled instances whose source queries come from Part I, and 150 sampled instances whose source queries come from Part II. Because these two subsets are constructed differently, we use two closely related scoring schemes on a $[-3, 3]$ scale. In both cases, higher scores indicate stronger validity, while negative scores indicate clear mismatch or irrelevance. We built a simple annotation platform to support this process, and detailed annotation criteria and interface examples are provided in our code repository. All annotations were completed by a single annotator, and each sampled instance was labeled once.

For each sampled query–agent pair, the annotator judges whether the synthesized configuration is a reasonable capability match to the user request, based on the backbone model, tool composition, and the functional requirements expressed in the query. We also inspect the constituent model and tool components, so that we can distinguish overall agent-level plausibility from component-level correctness. For Part I-derived queries, the backbone LLM is already labeled in the source benchmark and is therefore treated as valid by construction; the main focus of verification is the additional tool augmentation on top of the original LLM-only agent. For Part II-derived queries, the labeled toolset is treated as valid by construction, while the verification focuses on the additionally retrieved tools and the expanded backbone LLM.

Table 8 shows that the synthesized labels in Part III are not arbitrary pseudo-positives, although they are clearly not noise-free. For Part I-derived samples, the newly added retrieved tools are often noisy at the component level, but the final synthesized agents still receive a high average human score of 2.74/3, with 96.71% rated as 2 or 3. For Part II-derived samples, the tool side is substantially stronger while the expanded backbone LLM is only weakly validated; even so, the final agents achieve an average score of 2.12/3, with 79.87% rated as 2 or 3. Overall, these results suggest that Part III preserves substantial human-recognizable capability-matching signal despite structured synthesis noise, supporting its use as a practical large-scale supervision source for benchmark construction.

*Table 8.* Summary of human verification results for Part III synthesized labels on sampled subsets derived from Part I and Part II. The two subsets follow different construction routes: Part I-derived instances inherit the backbone LLM as valid by construction and mainly evaluate tool augmentation, while Part II-derived instances inherit the labeled toolset and mainly evaluate additional tool retrieval and backbone LLM expansion. The results reveal distinct noise patterns across the two synthesis routes.

| Metric | Part I-derived | Part II-derived |
|---|---|---|
| Sampled records | 195 | 150 |
| Agent candidates inspected | 975 | 750 |
| LLM retrieval items inspected | 970 | 716 |
| Tool retrieval items inspected | 1499 | 2099 |
| Internal agent tool items inspected | 626 | 1460 |
| LLM score mean | 3.00 | 0.79 |
| LLM score = 3 (%) | 100.00 | 0.00 |
| Tool score mean | -1.72 | 1.35 |
| Tool score $\geq$ 2 (%) | 14.68 | 61.41 |
| Tool score = 3 (%) | 9.34 | 54.12 |
| Tool score < 0 (%) | 74.32 | 20.10 |
| Tool score = -3 (%) | 73.18 | 19.87 |
| Agent score mean | 2.74 | 2.12 |
| Agent score = 3 (%) | 77.33 | 43.47 |
| Agent score = 2 (%) | 19.38 | 36.40 |
| Agent score = 1 (%) | 3.28 | 13.60 |
| Agent with zero tools (%) | 53.64 | 8.13 |
| Agent with any negative internal tool (%) | 8.72 | 2.00 |
| Agent with all internal tools scored 3 (%) | 75.38 | 89.47 |
| Record with at least one score-3 retrieved tool (%) | – | 99.33 |
| Record with at least one negative retrieved tool (%) | – | 74.67 |
| Top-1 retrieved tool scored 3 (%) | – | 66.00 |

## H. Progressive Dialog Interactions Improve Agent Recommendation Performance

We study agent recommendation in a *narrative-driven recommendation* setting, where the user's needs, preferences, profile, and constraints are conveyed directly through a natural-language query, rather than through explicitly constructed user identifiers and their historical behaviors. Accordingly, our benchmark is user-ID-free: the recommender must infer the appropriate agent configuration from the query text itself, instead of exploiting explicit user identities, repeated interactions, or long-term preference traces as is common in conventional recommendation.

*Table 9.* Analysis of progressive interactions in agent recommendation. At step $k$, the query is formed by concatenating the first $k$ turns of the session.

| Step | Count | P@10 | R@10 | nDCG@10 | MRR@10 | Step | Count | P@10 | R@10 | nDCG@10 | MRR@10 |
|---|---|---|---|---|---|---|---|---|---|---|---|
| 1 | 15232 | 0.0919 | 0.9189 | 0.8482 | 0.8250 | 11 | 17 | 0.1000 | 1.0000 | 0.9489 | 0.9314 |
| 2 | 15232 | 0.0952 | 0.9522 | 0.8874 | 0.8662 | 12 | 12 | 0.1000 | 1.0000 | 0.9122 | 0.8854 |
| 3 | 89 | 0.0944 | 0.9438 | 0.8395 | 0.8055 | 13 | 11 | 0.1000 | 1.0000 | 0.9210 | 0.8939 |
| 4 | 79 | 0.0962 | 0.9620 | 0.8745 | 0.8449 | 14 | 11 | 0.1000 | 1.0000 | 0.9329 | 0.9091 |
| 5 | 61 | 0.0984 | 0.9836 | 0.8775 | 0.8415 | 15 | 10 | 0.1000 | 1.0000 | 0.9631 | 0.9500 |
| 6 | 52 | 0.0981 | 0.9808 | 0.8911 | 0.8606 | 16 | 8 | 0.1000 | 1.0000 | 0.9539 | 0.9375 |
| 7 | 44 | 0.0977 | 0.9773 | 0.9186 | 0.8977 | 17 | 6 | 0.1000 | 1.0000 | 0.9385 | 0.9167 |
| 8 | 36 | 0.1000 | 1.0000 | 0.9405 | 0.9206 | 18 | 5 | 0.1000 | 1.0000 | 0.9262 | 0.9000 |
| 9 | 30 | 0.1000 | 1.0000 | 0.9275 | 0.9037 | 19 | 2 | 0.1000 | 1.0000 | 1.0000 | 1.0000 |
| 10 | 21 | 0.1000 | 1.0000 | 0.9491 | 0.9325 | 20 | 1 | 0.1000 | 1.0000 | 1.0000 | 1.0000 |

Importantly, the absence of user IDs does not imply that the benchmark is limited to strictly single-shot inputs. While identity-level signals are unavailable, many queries in our dataset still carry session-level contextual information. In particular, Part II and its derived Part III contain multi-turn conversational context accumulated across dialogue turns. This context is often useful for recommendation, since later turns may refine user requirements, add constraints, or clarify the desired functionality. To better understand the role of such accumulated context, we conduct a progressive query expansion analysis on Part II.

For each dialogue session with turns $(Q_1, Q_2, \ldots, Q_t)$, we construct a sequence of recommendation instances by cumulatively concatenating the observed turns, namely $(Q_1)$, $(Q_1 + Q_2)$, $(Q_1 + Q_2 + Q_3)$, and so on. We then evaluate the same recommender at each step and measure how ranking quality changes as more conversational context becomes available.

Table 9 summarizes the results. The clearest improvement evidence comes from the transition between Steps 1 and 2 show that even limited additional dialogue context can strengthen capability matching. For longer sessions, although the number of samples becomes much smaller, the overall trend remains positive, with ranking quality continuing to improve, evidenced by nDCG and MRR. Meanwhile, the sharp reduction in sample count after Step 2 suggests that, in most agent recommendation cases, the suitable agent is already identifiable within the first few turns. This highlights a key property of our setting: multi-turn context is helpful, but the task is typically short-horizon rather than requiring long interaction chains.

