# OpenReview forum: "AgentSelect: Benchmark for Narrative Query-to-Agent Recommendation"
_ICML.cc/2026/Conference — ICML 2026 regular_

### Official Review · Reviewer_jeQ7 · 2026-03-12

**Soundness:** 2
**Presentation:** 2
**Significance:** 3
**Originality:** 3
**Overall Recommendation:** 4
**Confidence:** 4

**Summary:**

This paper introduces AgentSelect, a benchmark for ranking deployable agent configurations (backbone model $M$ and toolset $T$) based on free-form narrative queries. The authors unify heterogeneous evaluation data from over 40 sources into a standardized, positive-only interaction dataset comprising 111,179 queries, 107,721 agents and 251,103 interaction records. The benchmark is partitioned into three parts: LLM-only (Part I), toolkit-only (Part II), and synthesized compositional agents (Part III).

**Compliance With Llm Reviewing Policy:**

Affirmed.

**Final Justification:**

The rebuttal answered and clarified my questions, I keep my original positive score.

**Key Questions For Authors:**

1. For the synthesized compositional agents, how do we evaluate the quality?
2. Are there training/testing splitting for results reported in Table 2?

**Limitations:**

The representation of an agent may be too oversimplified.

**Strengths And Weaknesses:**

Strengths
1. As agent marketplaces grow, the task of on-demand agent selection becomes important. This study addresses this gap to provide the first unified data and evaluation infrastructure with over 100k agents from 40+ sources for agent recommendation.

2. The study transform the agent selection to be a learning to rank problem, model each agent as (LLM model $M$ and toolset $T$), and rank relevant agents for an input natural language query $Q$, so that the canonical item recommendation methods can be used for agent retrieval.
3. The authors conduct a comprehensive empirical evaluation of 18 representative algorithms across six recommendation families on the AgentSelect benchmark, to establish a reproducible baseline for narrative agent retrieval.

Weaknesses
1. Agent representation: this study represents an agent by its backbone language model $(M)$ and toolset $(T)$. This assumption may be oversimplified as two agents with the same $M$ and $T$ but different prompting strategies or memory architectures can have very different outcomes.
2. This study assumes the user's intent is fully captured in a single query $Q$ with no session history. This ignores the iterative nature of real agent interactions where the "best" agent might only be identifiable after several turns.
3. Experiment setup details are not clear, are there training/testing splitting for results reported in Table 2?
4. The writing can be improved, some minor issues:
(a) $(M, T)$ first occurred in Abstract without explanation; there is no explanation of $C$ for $(M, T, C)$
(b) Acknowledgements section shall be removed

---

> ### Author Rebuttal · Authors · 2026-03-30
>
> We sincerely appreciate your positive assessment that our work addresses an emerging and practically important problem in the agent ecosystem. We are encouraged that the reviewer recognized the value of framing agent selection as a ranking problem and establishing a unified benchmark with broad empirical comparisons. Below we respond to the main concerns and questions.
>
> **Summarized W1:** Agent representation by ((M,T)) may be oversimplified.
>
> **A->W1:** Thank you for this important concern. We agree that a fully deployed agent depends on more than its backbone model and toolset. Prompting strategies, memory, policies, and runtime parameters can also affect behavior. However, modeling all these factors would require a much richer formulation and substantially stronger supervision than is currently available in the emerging agent ecosystem. Our benchmark therefore intentionally adopts ((M,T)) as a **capability-level profile**: the backbone largely determines reasoning/generation capacity, while the toolset determines external action ability. In this sense, the representation is not intended to capture every implementation detail, but to characterize what an agent is functionally able to do. We will clarify this limitation more explicitly in the revision.
>
> **Summarized W2:** The benchmark ignores session history and may miss the iterative nature of realistic agent interaction.
>
> **A->W2:** Thank you for this helpful point. We agree that many real-world agent interactions are iterative. **Part II and many cases in Part III in AGENTSELECT does have session-level interaction information, which is modeled as contextual information in a given query.** We will clarify this point in our Section 3. We also added **Appendix H** to further show that progressively incorporating dialogue context consistently improves recommendation performance.
>
> **Summarized W3 & Q2:** Experimental setup details, especially train/test split for Table 2, are unclear.
>
> **A->W3 & Q2:** Thank you for pointing this out. Yes, the results in **Table 2** are obtained under a standard 8:2 train/test split, and we agree that this should have been stated more clearly. We have revised the manuscript to explicitly describe the data split and evaluation protocol in the Section 5.
>
> **Summarized Q1:** How is the quality of synthesized compositional agents evaluated?
>
> **A->Q1:** Thank you for this important question. We agree that the quality of Part III synthesized agents should be carefully examined. To address this, we added **Appendix G: Reliability Analysis of Part III Synthesized Labels via Human Verification**. The results show that although the retrieval stage and LLM judgment are not noise-free, the final synthesized agents remain broadly consistent with human judgment. For the Part I-derived subset, synthesized agents achieve an average human score of **2.74/3**, with **96.71%** rated as 2 or 3. For the Part II-derived subset, the average score is **2.12/3**, with **79.87%** rated as 2 or 3. These results suggest that Part III is not formed by simply treating retrieved components as positives; rather, noisy retrieved models/tools can be filtered during the synthesis stage.
>
> **Summarized W4:** Writing and presentation can be improved.
>
> **A->W4:** Thank you for these careful suggestions. We agree that the writing can be improved. In the revision, we will (1) explain ((M,T)) clearly at its first appearance, including the meaning of (C) in ((M,T,C)), and (2) remove the Acknowledgements section to comply with the submission policy. We will also further polish the presentation for clarity and consistency throughout the paper.

---

> > ### Author Rebuttal · Reviewer_jeQ7 · 2026-04-04
> >
> > Thanks for the explanations and additional analysis. I keep my original positive score.

---

> > > ### Author Response · Authors · 2026-04-04
> > >
> > > Thanks for supporting our work! We will incorporate these explanations and analysis into our revision.

---

### Official Review · Reviewer_XarN · 2026-03-12

**Soundness:** 3
**Presentation:** 3
**Significance:** 3
**Originality:** 2
**Overall Recommendation:** 4
**Confidence:** 3

**Summary:**

This paper introduces AgentSelect, a benchmark for narrative query-to-agent recommendation. The authors argue that when user needs are expressed as natural-language narratives, existing collaborative filtering, graph neural network, and traditional matching-based methods struggle to capture the semantic alignment between user intent and agent capabilities. To address this gap, the paper presents a benchmark that includes a task formulation, a dataset construction pipeline, an evaluation protocol, and a set of baseline comparisons. The empirical study compares retrieval-based and generative approaches, and the results suggest that LLM-based semantic reasoning methods substantially outperform traditional recommendation approaches in this setting, pointing to a potentially important shift in how recommendation should be approached in narrative agent-selection scenarios.

**Compliance With Llm Reviewing Policy:**

Affirmed.

**Final Justification:**

In light of my previous comments and the authors’ rebuttal, I have decided to maintain my original score and continue to recommend acceptance.

**Key Questions For Authors:**

1. In Part III, candidate configurations are validated by an LLM judge. How was the judge’s reliability assessed, for example via comparison with human annotations or execution outcomes on a subset? It would also be helpful to discuss how judge bias might affect the benchmark.
2. The data generation process needs further clarification. Are narrative queries written by humans, collected from real users, generated by LLMs, or mixed across sources? If mixed, what are the proportions, and is there any risk of distribution leakage?
3. The one-off setting should be defined more precisely. Are historical interaction signals completely removed, and have the authors considered hybrid settings that combine limited history with narrative queries? This would help interpret the failure of CF- and GNN-based methods.
4. Have the authors evaluated cross-domain transfer or zero-shot generalization to unseen agent pools? Such results would help assess the robustness of the proposed benchmark and methods.

**Limitations:**

yes

**Strengths And Weaknesses:**

Strengths：
S1: The paper studies a forward-looking and practically relevant problem. Narrative query-to-agent recommendation is increasingly important with the rise of LLM agents, agent marketplaces, tool-use systems, and multi-agent orchestration frameworks. This setting differs meaningfully from conventional item recommendation and is well motivated from both practical and research perspectives.
S2: The paper provides a clear task definition, a reasonably transparent data construction pipeline, and a well-scoped evaluation protocol. This makes the work potentially useful as a shared testbed for future research in this emerging area.
S3:  The empirical study provides useful evidence that traditional recommendation paradigms degrade substantially in one-off recommendation settings without rich historical interaction signals. This observation is informative for the recommendation community and helps clarify the limitations of collaborative methods in narrative, intent-driven scenarios.
S4: The comparison across different modeling paradigms is timely and valuable. In particular, contrasting representation-based retrieval methods with generative reasoning-based approaches helps illuminate what kinds of inductive biases are more suitable for this task.
Weaknesses：
W1: The methodological novelty is limited. The main contribution is the construction of the benchmark and the accompanying empirical analysis, rather than a fundamentally new modeling framework, learning objective, or optimization strategy. Most evaluated methods appear to be adaptations or fine-tuned variants of existing approaches.
W2: There is potential bias in the synthesized positive examples. In Part III of the benchmark, candidate configurations are validated using an LLM judge after retrieval-based composition. While this is a pragmatic design choice, the use of LLM-as-a-judge may introduce bias and may not fully reflect true task-execution success. The paper does provide some supporting evidence through counterfactual tests and end-to-end validation, but a more thorough analysis of judge reliability, calibration, or agreement would strengthen confidence in the benchmark.
W3: The data construction process may introduce distributional bias or annotation artifacts. If narrative queries are partially generated or rewritten by annotators or LLMs, the resulting benchmark may not fully reflect the diversity and messiness of real user requests. A more detailed analysis of data authenticity, linguistic diversity, and possible construction artifacts would improve the external validity of the benchmark.
W4: The evaluation metrics appear largely conventional. If the study primarily relies on ranking-based metrics such as Recall@K or NDCG@K, these may not fully capture functional correctness or downstream task success in agent recommendation settings. Additional execution-level, tool-composition-level, or task-success-oriented evaluations would make the benchmark substantially stronger.
W5: While the empirical results clearly show degradation of CF- and GNN-based approaches, the paper does not provide a sufficiently principled explanation for this failure mode. For instance, in one-off settings, the lack of repeated interactions may undermine the assumptions behind collaborative factorization, while sparse or weakly informative graphs may limit the effectiveness of message passing. Even a semi-formal analysis along these lines would strengthen the paper’s central claim.

---

> ### Author Rebuttal · Authors · 2026-03-30
>
> We sincerely appreciate your positive assessment of our paper. We are encouraged that you recognized the clear task formulation, transparent benchmark design, and empirical value of our study. Below we address your concerns concisely.
>
> **Summarized W1:** Methodological novelty is limited
>
> **A->W1:** Thank you. We agree that our main contribution is the benchmark—its formulation, construction, and evaluation protocol—rather than a new recommendation algorithm. We believe such a shared testbed is timely and valuable for this emerging setting, while stronger task-specific models are an important direction for future work.
>
> **Summarized W2&Q1:** Bias concern about Part III synthesized agents, and suggestion for adding reliability analysis.
>
> **A->W2&Q1:** Thank you. We agree that LLM-based synthesis may introduce bias. To examine this, we added **Appendix G: Reliability Analysis of Part III Synthesized Labels via Human Verification**. The results show that although retrieval and LLM judgment are not noise-free, the final synthesized agents remain broadly consistent with human judgment. For Part I-derived samples, retrieved tools are often noisy, yet synthesized agents still achieve **2.74/3** average human score, with **96.71%** rated 2 or 3. For Part II-derived samples, the expanded backbone LLM side is less reliable, but synthesized agents still achieve **2.12/3**, with **79.87%** rated 2 or 3.
>
> **Summarized W3&Q2:** Concern about query authenticity, distributional bias, and construction artifacts.
>
> **A->W3&Q2:** Thank you. AGENTSELECT does not rewrite the original narrative queries; we preserve them as collected from the 40 source datasets, retaining their natural diversity and noise. Part I.2 also uses sampling only, without paraphrasing. We further conducted a lightweight corpus analysis on query length, lexical diversity, and simple duplicate/template patterns. The corpus mainly consists of substantial narrative requests (**median 78 words; p25=50, p75=126**) with reasonably high lexical diversity (**mean TTR 0.698**). We do observe some repeated normalized forms, which is expected because part of the benchmark comes from structured evaluation datasets such as **MATH, BBH, and MMLU**, where instruction patterns naturally overlap.
>
> **Summarized W4:** Suggestion of reflecting real task success.
>
> **A->W4:** Thank you. We agree that ranking metrics alone do not fully capture end-to-end agent performance. However, AgentSelect is defined as a **query-to-agent recommendation benchmark**, where the primary goal is to rank candidate agents by capability match rather than directly optimize execution outcomes, which also depend on tool quality, API stability, and runtime variance. To partially address this, **Appendix B (Table 7)** provides a deployment-centric validation showing that higher-ranked agents tend to achieve better judged performance after actual instantiation and execution. Broader execution-level evaluation is important, but beyond the current scope.
>
> **Summarized W5:** Add failure mechanism analysis of CF/GNN methods
>
> **A->W5:** Thank you. We agree this mechanism should be stated more explicitly. In our setting, one-off queries weaken the repeated co-preference assumption behind CF, while sparse and weakly informative interaction graphs limit effective message passing for GNNs. Our analyses in Sections 6 and 7 already support this interpretation, and we will make it clearer in the revision.
>
> **Summarized Q3:** Clarification on one-off narrative recommendation setting and whether limited-history are considered.
>
> **A->Q3:** Thank you. In AGENTSELECT, the one-off setting means that we do not model persistent user identity or cross-session history, so historical interaction is not used as an explicit personalization source. However, **Part II and many Part III cases do contain session-level context**. We also added **Appendix H**, which shows that progressively incorporating dialogue context consistently improves recommendation performance. More generally, this setting can naturally extend to richer history, e.g., when long-term information is summarized by a memory module and injected into the query.
>
> **Summarized Q4:** Generalization to unseen agent
>
> **A->Q4:** Thank you. Yes—**Section 9.1** is exactly designed to test generalization to an unseen agent pool. In that experiment, models trained on AGENTSELECT are evaluated on the **MuleRun** marketplace, whose agent catalog is entirely disjoint from the training pool. The positive transfer results suggest that the benchmark captures transferable query-to-agent matching signals. We also updated the prompting templates and generated queries in the anonymous repository.

---

> > ### Author Rebuttal · Reviewer_XarN · 2026-04-04
> >
> > Thank you to the authors for the careful rebuttal. I have no further questions.

---

> > > ### Author Response · Authors · 2026-04-04
> > >
> > > Very glad our rebuttal can solve all your concerns!

---

### Official Review · Reviewer_ccuz · 2026-03-12

**Soundness:** 3
**Presentation:** 2
**Significance:** 2
**Originality:** 3
**Overall Recommendation:** 4
**Confidence:** 3

**Summary:**

The authors attempt to investigate a notable area in the emerging LLM agent ecosystem by introducing AGENTSELECT, a benchmark for narrative query-to-agent recommendation. A central aspect examined by this manuscript is the construction of a unified dataset that converts heterogeneous evaluation artifacts into query–agent interaction data. Experiments reveal a long-tail distribution of agent utility and show that content-aware matching methods outperform ID-based collaborative filtering, with additional validation through transfer to a public agent marketplace and deployed agent execution scenarios.

**Compliance With Llm Reviewing Policy:**

Affirmed.

**Final Justification:**

The rebuttal addressed my main concerns, and I  increased my Overall Recommendation from 3 to 4.

**Key Questions For Authors:**

1.	The evaluation mainly focuses on traditional recommendation algorithms. Considering the semantic-heavy nature of the AGENTSELECT benchmark, comparisons with agent-based recommendation approaches that leverage semantic understanding or reasoning capabilities are currently missing. Relevant approaches such as those discussed in [1] and [2] could provide useful reference points for evaluating the benchmark.
2.	The paper lacks a concrete real-world example illustrating how a system dynamically recommends a specific (Model, Tool) combination for a single narrative query, which would help clarify the practical usage scenario of the proposed benchmark.

[1] Zhang A, Chen Y, Sheng L, et al. On generative agents in recommendation[C]//Proceedings of the 47th international ACM SIGIR conference on research and development in Information Retrieval. 2024: 1807-1817.

[2] Xu W, Shi Y, Liang Z, et al. iAgent: LLM agent as a shield between user and recommender systems[C]//Findings of the Association for Computational Linguistics: ACL 2025. 2025: 18056-18084.

**Limitations:**

Yes. However, several additional limitations remain:

The benchmark mainly evaluates ranking performance using metrics such as nDCG and MRR, while practical execution factors are not considered. Aspects such as API success rates, latency, and operational costs are not included. As a result, highly ranked agents may not necessarily be functional or efficient in real-world deployments.

The paper does not discuss potential privacy and security risks associated with recommending third-party tools or agent configurations. Issues such as prompt injection, sensitive data exposure, and security vulnerabilities in external tools are not addressed.

**Strengths And Weaknesses:**

Strength：
1. The paper formalizes an emerging but previously under-defined task, namely query-conditioned recommendation of end-to-end agents rather than selecting only a backbone LLM or a single tool. The capability-profile abstraction together with the YAML-based representation provides a clear and concrete formulation for deployable agents.
2. The paper constructs a large-scale benchmark containing 111,179 narrative queries, 107,721 deployable agents, and 251,103 positive interaction records. The dataset covers three practical scenarios including LLM-only agents, toolkit-only agents, and compositional agents, which helps fill an important data gap for end-to-end agent recommendation research.
3. The paper evaluates a wide range of recommendation baselines and empirically shows that traditional collaborative filtering and ID-based methods perform poorly under the long-tail interaction regime. These results highlight the difficulty of the proposed task and demonstrate that AGENTSELECT serves as a challenging benchmark for agent recommendation research.




Weakness:

1.	This paper risks circular dependency and bias propagation in its Part III synthesis pipeline. By relying on pre-trained retrievers and an LLM judge to generate ground-truth labels, the final model may simply inherit their inherent biases rather than learning genuine agent utility. Furthermore, the supervision's reliability is insufficiently justified, as the paper lacks both an analysis of the LLM judge's bias and a direct quantitative comparison against realistic alternatives.

2.	This paper lacks justification for framing its task as "recommendation." By focusing solely on session-based queries without persistent user history, the setup is fundamentally a stateless routing problem. It fails to distinguish this approach from existing LLM routing frameworks, making the recommendation terminology seem unmotivated.

3. This paper generates candidate configurations in Part III by taking the Cartesian product of retrieved LLM and tool shortlists. However, this design assumes perfect orthogonal composability and overlooks real-world compatibility constraints. In practice, specific models often require distinct API schemas or formats, meaning many of these blindly paired configurations may be functionally incompatible or illogical during actual execution.

4. MuleRun evaluation in Table 6 uses 20 handcrafted requests per agent, while the Agno deployment study in Appendix B includes only 200 queries with five candidates per query, which may not fully stress-test robustness under diverse and noisy user queries.

5. Figure 2 exhibits visual anomalies suggesting it may be AI-generated (for example, the inequality symbol in the bottom right corner is rendered discontinuously). Furthermore, several elements within the figure are ambiguous; specifically, the intended meaning of the icons in the bottom row is not clearly explained. Lastly, some text within the figure is difficult to read, and the differing font styles result in a lack of visual consistency with Figure 1.

---

> ### Author Rebuttal · Authors · 2026-03-30
>
> We sincerely thank you for the careful assessment and constructive suggestions. We appreciate the recognition of the task importance, benchmark scale, and the practical and technical challenges of agent recommendation. Below we respond briefly to each point.
>
> **Summarized W1:** Part III bias and reliability.
>
> **A->W1:** Thank you for raising this important concern. We agree that Part III may inherit bias from both retrieval and LLM judging. To assess reliability, we added **Appendix G: Reliability Analysis of Part III Synthesized Labels via Human Verification**. The results show that, while not noise-free, the synthesized labels are broadly consistent with human judgments. For the Part I-derived subset, synthesized agents achieve an average human score of **2.74/3**, with **96.71%** rated 2 or 3. For the Part II-derived subset, the average score is **2.12/3**, with **79.87%** rated 2 or 3. Additional details are provided in the anonymous repository.
>
> **Summarized W2:** Why this task is framed as recommendation.
>
> **A->W2:** Thank you for this important concern. AGENTSELECT does not model persistent identity or cross-session history, but this does not make it non-recommendation. Prior narrative-driven (OCG-Agent), instruction-aware (iAgent, PersonaX), and content-aware (EasyRec) recommendation settings also rely on intent and preference signals expressed directly in the query or summarized into it. Our setting is therefore best viewed as **session-level recommendation**. We also added **Appendix H**, showing that adding dialogue context consistently improves performance. By contrast, routing typically selects among a small set of endpoints under quality-cost trade-offs, whereas our task ranks a large agent set using semantic, tool, and collaborative signals, which is structurally closer to recommendation.
>
> **Summarized W3 & Limitations:** Compatibility, validity, and applicability of Part III agents.
>
> **A->W3:** Thank you for raising this point. We do not view compatibility as a fundamental obstacle. In practice, it is increasingly handled by the ecosystem layer: frameworks such as Agno and LangChain already support heterogeneous backbones, tool wrappers, and multi-model collaboration, and non-tool-capable models can also be incorporated through proxy or agentic reasoning pipelines. Our Agno-based implementation (Appx. B, D) follows this abstraction, and emerging protocols such as MCP and Agent Skills further support this trend.
>
> For executability, neither the benchmark nor Part III is intended to certify universal runnability. Requiring every synthesized agent to be directly executable is unrealistic in a fast-evolving ecosystem. The key question is instead whether the benchmark provides a meaningful simulated testbed and whether its supervision supports transferable query-to-agent matching. This is exactly what **Sec. 9.1** (transfer) and **Sec. 9.2** (execution-oriented validation) are designed to evaluate. We will clarify this more explicitly in the revision.
>
> **Summarized W4:** Concern about query diversity.
>
> **A->W4:** Thank you for your concern. In MuleRun, queries are not simple handcrafted prompts; they are constructed with controlled difficulty and noise by considering both the target agent and functionally similar distractor agents. This makes the evaluation a more realistic test of fine-grained capability discrimination. We also update the prompting templates and generated queries in the anonymous repository.
>
> For Agno, we are expanding the evaluation to 500 queries. This is a fully deployed end-to-end evaluation: each run takes about **70 hours**, and we report the average over 3 runs. At this cost, 500 queries is already a reasonably large scale; for reference, SWE-bench also uses 500 instances. We will update the expanded results in the anonymous repository.
>
> **Summarized W5:** Improvement of Figure 2.
>
> **A->W5:** Thank you for this careful suggestion. We apologize that Figure 2 was not clear enough. AI assistance was used only for icon drafting. The bottom row is intended to illustrate the implicit-feedback assumption in recommendation.
>
> **Summarized Q1:** Suggestion to add Agent4Rec and iAgent.
>
> **A->Q1:** Thank you for this valuable suggestion. We did but omitted them due to page limits. They are now added in the anonymous repository.
>
> **Summarized Q2:** Need a clearer real-world usage example.
>
> **A->Q2:** Thank you for this suggestion. We already include examples in the Introduction, Appendix D, Figure 5, and the demo. We will clarify two direct scenarios: **agent marketplaces**, where the goal is to select existing agents (e.g., MuleRun), and **orchestration systems**, where the goal is to compose and rank candidate agents (e.g., Chatbox, Agno).

---

> > ### Author Rebuttal · Reviewer_ccuz · 2026-04-03
> >
> > I thank the authors for providing the additional empirical results. In light of these efforts, I will increase my rating from 3 to 4.
> >
> > I encourage the authors to incorporate the new experimental results and clarifications from the rebuttal into the revised manuscript, and to revise the presentation of the figure to make it clearer and easier to understand.

---

> > > ### Author Response · Authors · 2026-04-04
> > >
> > > Hi Reviewer ccuz, thank you for raising your score. We really appreciate your helpful feedback and support in improving our paper, and we will revise it accordingly.

---

### Official Review · Reviewer_1rTH · 2026-03-13

**Soundness:** 4
**Presentation:** 4
**Significance:** 4
**Originality:** 3
**Overall Recommendation:** 5
**Confidence:** 4

**Summary:**

This paper introduces AgentSelect, a benchmark for narrative query-to-agent recommendation that converts heterogeneous evaluation artifacts (LLM leaderboards, tool-use benchmarks, and agent evaluations) into a unified query-agent interaction dataset. A central aspect examined by the paper is the formulation of agent selection as a recommendation and ranking problem over capability profiles defined by backbone models and toolsets. The benchmark aggregates large-scale data across multiple sources and evaluates several recommender architectures to study how models align user queries with agent configurations. Experimental results suggest that content-aware matching approaches perform better than traditional collaborative filtering methods under sparse interaction regimes.

**Compliance With Llm Reviewing Policy:**

Affirmed.

**Final Justification:**

The paper is technically solid, well-motivated, and addresses a timely problem with strong empirical evaluation. Strengths include the novel benchmark, comprehensive experiments, and practical relevance. Weaknesses mainly concern assumptions in data synthesis and simplified agent representation. The rebuttal adequately addressed my concerns, particularly with added validation and clarifications. Overall, my assessment remains unchanged, and I recommend acceptance.

**Key Questions For Authors:**

1. How robust is the synthesized compositional interaction signal (Part III) to errors in component retrieval or assumptions about which tools and models satisfy a given query?

2. The agent capability representation focuses on backbone models and toolsets. How do the authors justify excluding other factors such as prompts, policies, or memory mechanisms that can significantly affect agent behavior?

3. In the real-world validation experiments, how representative are the selected tasks and agents of practical agent marketplaces, and how sensitive are the results to the specific evaluation setup?

4. To what extent does the benchmark capture realistic user interactions in agent marketplaces, especially given that it assumes queries without persistent user identity or interaction history?

**Limitations:**

Yes.

**Strengths And Weaknesses:**

**Strengths**

(+) The paper addresses an emerging and practically relevant problem of selecting appropriate LLM agents for user queries within large agent catalogs and marketplaces, which reflects a realistic deployment scenario.

(+) The proposed benchmark aggregates heterogeneous sources such as LLM evaluation leaderboards and tool-use benchmarks into a unified recommendation-style dataset, potentially enabling new research directions at the intersection of agent systems and recommender systems.

(+) The authors conduct a broad experimental evaluation covering multiple recommender architectures and analyze phenomena such as long-tail sparsity and the shift from popularity-based methods to content-aware matching.

(+) The paper includes additional analyses such as modality ablations and counterfactual capability edits to examine how recommendation models respond to changes in agent capabilities.

---

**Weaknesses**

(-) The synthesis process used to generate compositional interactions (Part III) relies on retrieved components and pseudo-positive signals, which may introduce biases or assumptions about capability relevance that are not fully validated.

(-) Some aspects of the benchmark design, such as the representation of agents solely through backbone models and toolsets, may oversimplify real agent configurations where prompting strategies, policies, or runtime parameters also influence performance.

(-) The paper could better clarify how tools and model components interact within the capability profile abstraction and how this representation captures meaningful differences between agent configurations.

---

> ### Author Rebuttal · Authors · 2026-03-30
>
> We sincerely appreciate your positive assessment that our work addresses an emerging practical problem in the agent ecosystem. We are particularly encouraged that you recognized the benchmark’s potential to stimulate new research in the intersection of agent systems and recommender systems. We also thank you for acknowledging our broad experimental evaluation and in-depth analyses of sparsity, modality effects, and capability sensitivity. Your feedback is highly valuable, and we provide detailed responses to your concerns below.
>
> **Summarized W1&Q1:** The PartIII data synthesis process may introduce invalid or biased capability relevance. The robustness concern of PartIII's data to retrieval noise.
>
> **A->W1&Q1:** Thank you for this important concern. To address it, we added a new **Appendix G: Reliability Analysis of Part III Synthesized Labels via Human Verification**. The results show that although the synthesized labels exhibit little extent bias but it is broadly consistent with human judgments. While the retrieval process is not entirely noise-free, the noise does not substantially carry to the final synthesized agents.
>
> - Specifically, for the Part I-derived subset, retrieved tools are often noisy: 74.32% of retrieved tools are noisy. However, the final synthesized agents still achieve a high average human score of 2.74/3, with 96.71% rated as 2 or 3.
> - For the Part II-derived subset, the tool side is substantially stronger while the expanded backbone LLM is more weakly validated and can still be noisy. Even so, the final synthesized agents achieve an average human score of 2.12/3, with 79.87% rated as 2 or 3.
>
> These results support two points. First, Part III is not constructed by directly treating retrieved components as positives. Second, noisy retrieved tools or models can be filtered during synthesis by the LLM judge. We also release the synthesis prompt in the anonymized repository for transparency.
>
>
>
> **Summarized W2&Q2:** Concern that representing agents only by backbone models and toolsets may oversimplify
>
> **A->W2&Q2:** Thank you for this important practical concern. We agree that a fully deployed agent depends on more than its backbone model and toolset. Prompting strategies, policies, runtime parameters, and memory can also influence agent performance. However, modeling these factors would require a much richer formulation and substantially stronger supervision than what is currently available at this stage of the agent ecosystem. As such, our formulation is intentionally at the level of a capability profile.  We believe this level of investigation is better left to future work as the ecosystem and corresponding supervision resources mature.
>
> **Summarized W3:** Clarify tools and models' interactions
>
> **A->W3:** Thank you for this helpful suggestion for stating more explicitly about the interaction between the backbone model and tools. In our formulation, (M,T) jointly determine the functions the agent can perform. At inference time, the backbone model invokes tools when needed; when a backbone is not directly tool-capable, tool use can still be mediated through an agentic reasoning pipeline. Thus, the profile is intended to capture the agent’s **capability-level functionality**.
>
> For the second point, our representation is designed to capture **meaningful functional differences** between agent configurations: changing the backbone (M) alters reasoning/generation capacity, while changing the toolset (T) alters external actions ability. In this sense, (M,T) is analogous to an electronic product profile that describes what it can do, rather than fine-grained configuration parameter.
>
>
> **Summarized Q3:** The representativeness and setup sensitivity of the real-world validation experiments.
>
> **A->Q3:** Thank you for this important question. We agree that these real-world experiments are supplementary, not exhaustive. Their purpose is to provide external validation from two complementary angles: transfer to an unseen public marketplace and consistency between recommendation ranking and deployed end-to-end performance. About sensitivity, the results may vary with the evaluation setup, but we think the overall trend shoud remain consistent.
>
> **Summarized Q4:** Realistic interactions concern
>
> **A->Q4:** Thank you for this important question. We agree that AGENTSELECT does not model persistent user identity and cross-session history. Our benchmark instead focuses on **session-level agent selection: when available, interaction history within a session is incorporated into the query as context.** This makes the setting particularly realistic for cold-start, session-bounded agent recommendation, which is common in public agent marketplaces and on-demand agent creation. We also add Appendix H, which shows that adding progressive dialog context consistently improves recommendation quality, supporting the practical value of session-level interaction modeling in our benchmark.

---

> > ### Author Rebuttal · Reviewer_1rTH · 2026-04-03
> >
> > Thank you for the clear and detailed rebuttal. Overall, my concerns are well addressed, and I will maintain my original score of 5.

---

> > > ### Author Response · Authors · 2026-04-04
> > >
> > > Thanks a lot for your in assessing our paper and view our response.

---

### Decision · Program_Chairs · 2026-04-30

**Decision:**

Accept (regular)

**Comment:**

This submission addresses a timely and practically important problem in the emerging LLM agent ecosystem: how to recommend an appropriate deployable agent configuration for a free form user query. Its main contribution is not a new learning algorithm, but a well-motivated and substantial benchmark that unifies heterogeneous evaluation artifacts into a large-scale narrative query-to-agent recommendation task, together with a broad empirical study across multiple recommendation paradigms. The paper is technically solid for its stated scope, and the experiments provide useful evidence that in long-tail, near one-off interaction regimes, content-aware semantic matching is substantially more effective than traditional ID based collaborative approaches. The reviewers appropriately noted limitations, especially the reliance on synthesized supervision in Part III, the simplified agent abstraction, and the still limited execution level validation; however, the rebuttal addressed these concerns in a satisfactory way by adding human verification, clarifying the intended capability-level scope of the representation, and strengthening external validation. Overall, I find the paper to make a meaningful benchmark and problem-formulation contribution that is likely to support further work in this area, and I recommend acceptance.